# DNA damage during S-phase mediates the proliferation-quiescence decision in the subsequent G1 via p21 expression

Alexis R. Barr[1,*], Samuel Cooper[1,2,*], Frank S. Heldt[3,*], Francesca Butera[1], Henriette Stoy[1], Jörg Mansfeld[4], Béla Novák[3] & Chris Bakal[1]

Following DNA damage caused by exogenous sources, such as ionizing radiation, the tumour suppressor p53 mediates cell cycle arrest via expression of the CDK inhibitor, p21. However, the role of p21 in maintaining genomic stability in the absence of exogenous DNA-damaging agents is unclear. Here, using live single-cell measurements of p21 protein in proliferating cultures, we show that naturally occurring DNA damage incurred over S-phase causes p53-dependent accumulation of p21 during mother G2- and daughter G1-phases. High p21 levels mediate G1 arrest via CDK inhibition, yet lower levels have no impact on G1 progression, and the ubiquitin ligases CRL4$^{Cdt2}$ and SCF$^{Skp2}$ couple to degrade p21 prior to the G1/S transition. Mathematical modelling reveals that a bistable switch, created by CRL4$^{Cdt2}$, promotes irreversible S-phase entry by keeping p21 levels low, preventing premature S-phase exit upon DNA damage. Thus, we characterize how p21 regulates the proliferation-quiescence decision to maintain genomic stability.

[1] Division of Cancer Biology, The Institute of Cancer Research, 237 Fulham Road, London SW3 6JB, UK. [2] Department of Computational Systems Medicine, Imperial College, South Kensington Campus, London SW7 2AZ, UK. [3] Department of Biochemistry, OCISB, University of Oxford, South Parks Road, Oxford OX1 3QU, UK. [4] Cell Cycle, Biotechnology Centre, TU Dresden, 01307 Dresden, Germany. * These authors contributed equally to this work. Correspondence and requests for materials should be addressed to A.R.B. (email: alexis.barr@icr.ac.uk) or to C.B. (email: chris.bakal@icr.ac.uk).

C ell cycle regulation balances the requirement of proliferation during tissue growth and homeostasis, with the need to ensure that damaged DNA is not propagated to future generations. Checkpoints have evolved to achieve control of cell cycle progression in response to DNA damage. The importance of DNA damage checkpoints is highlighted by the fact that their dysregulation is the fundamental basis of oncogenesis[1].

The tumour suppressor and transcriptional regulator p53 is stabilized in response to DNA damage and regulates the expression of numerous target genes involved in the DNA damage response[2,3]. Amongst the transcriptional targets of p53 is the Cyclin-dependent kinase (CDK) inhibitor p21 (refs 4,5). By inhibiting CDK activity, p21 mediates cell cycle arrest[6,7] downstream of p53, in response to DNA damage caused by exogenous sources, such as ionizing irradiation or chemical agents[5,8,9]. However, the role of p21 in cell cycle control in cells proliferating in the absence of such DNA damage is less clear. Moreover, whether p21 acts as a tumour suppressor *in vivo* is controversial[10]. For example, in p21 knockout (KO) mice exposed to exogenous DNA damage, both accelerated[11–14] and inhibited[15] tumourigenesis have been observed. However, the fact that p21KO mice develop spontaneous tumours[15], albeit later than p53-deficient mice[16], demonstrates that p21 plays an essential role in maintaining genomic stability.

Live measurements of p53 protein levels have shown that unperturbed cells exhibit transient pulses of p53 expression[17], suggesting that DNA damage is occurring in these cells. In response to transient p53 pulses, cells continue to cycle, yet more frequent pulses or sustained p53 signalling causes cells to arrest or undergo apoptosis[18]; thus mechanisms exist to temporally integrate DNA damage levels and p53 signalling[19,20]. Previous work in unperturbed cycling cells has also demonstrated that bifurcation in CDK2 activity after mitosis causes a subpopulation of cells to enter a p21-dependent G1 post-mitotic (pm) arrest state, prior to restriction point (RP) passage[21,22]. Taken together, we hypothesized that p53 signalling, in cells not exposed to exogenous DNA damage, could generate heterogeneity in p21 levels which, in turn, mediates the proliferation-quiescence decision in single cells. This would prevent the propagation of DNA damage to future generations and explain the importance of p21 in maintaining genomic stability.

Using a live single-cell imaging approach, we show that DNA damage accrued during S-phase, in non-transformed cycling cells, results in heterogeneity in p21 levels that is p53-dependent. We find that p21 accumulates during mother G2- and daughter G1-phases, where it modulates the proliferation-quiescence decision in daughter cells via CDK2 inhibition. Strikingly, we find that p21 accumulation below a threshold does not influence G1 progression. At these lower levels of p21, two ubiquitin ligases, CRL4$^{Cdt2}$ and SCF$^{Skp2}$, couple to remove p21 prior to the G1/S transition with markedly different rates. By combining single-cell measurements with mathematical modelling, we show that this p21 control system has the hallmarks of bistability, ensuring that the G1/S transition is irreversible. Our data therefore support a model in which p21 mediates a p53-dependent G1pm arrest in response to DNA damage occurring in S-phase.

## Results

**Cells exhibit cell-to-cell variability in p21 expression**. To quantify p21 protein levels over time in live cells, we introduced a GFP tag into the C-terminus of both alleles of the endogenous *CDKN1A* gene (Fig. 1a; Supplementary Fig. 1a,b). Gene-targeting was performed in an hTert-RPE1 cell line expressing one allele of

endogenously tagged mRuby-PCNA (proliferating cell nuclear antigen). This allowed us to accurately track cells and correlate changes in p21 with cell cycle phase (Fig. 1b; Supplementary Movie 1). p21-GFP was exclusively nuclear, matching the localization of endogenous p21 in hTert-RPE1 cells (Supplementary Fig. 1c). Population growth kinetics and cell cycle timing were unaffected by introduction of the GFP tag into p21, and p21-GFP is able to bind CDK2 (Supplementary Fig. 1d–f). Further confirmation that p21 function is retained by p21-GFP is that cells expressing p21-GFP are able to enter G1 arrest in high-serum growth conditions (Fig. 1f), whereas p21siRNA and p21KO cells are not (Fig. 2b).

Live cell tracking revealed marked cell-to-cell variability in p21-GFP levels amongst G1 and G2 hTert-RPE1 cells, within the same population (Fig. 1c). This variability was also seen in hTert-RPE1 cells fixed and immunostained for unlabelled p21 (Supplementary Fig. 2a). G1 cells have the highest and most variable p21-GFP levels, and there exists a strong correlation between p21-GFP levels and G1 phase length ($R = 0.62^{**}$; Fig. 1d). G2 is associated with lower and less variable p21-GFP levels and correlation between G2 length and p21-GFP levels is weaker than in G1 ($R = 0.51^{**}$; Supplementary Fig. 2b). Cells express undetectable levels of p21-GFP during S-phase (Fig. 1c), consistent with previous reports[23,24]. We do not observe p21-GFP degradation during mitosis, and mitotic p21-GFP expression does not affect mitotic timing (Supplementary Fig. 2c,d).

The lack of p21 degradation during mitosis would allow daughter cells to inherit p21 protein from their mother. Indeed, there is a positive correlation between p21-GFP levels in the mother cell G2 (G2$^M$) and daughter cell G1 (G1$^D$; $R = 0.75^{**}$; Supplementary Fig. 2e). Moreover, G1$^D$ p21-GFP levels in sister cells are positively correlated ($R = 0.81^{**}$), as well as G1$^D$ length between sisters ($R = 0.53^{**}$; Fig. 1e). Despite the strong correlation in p21-GFP levels in G2$^M$ with p21-GFP levels in G1$^D$, the phase lengths of G2$^M$ and G1$^D$ correlate poorly ($R = 0.07$, $P = 0.36$ Pearson's Correlation). Instead, we find G1$^D$ p21-GFP levels correlate better with combined G2$^M$ + G1$^D$ length ($R = 0.68^{**}$) than with G1$^D$ length alone ($R = 0.62^{**}$) suggesting that G2$^M$ length is not predictive of G1$^D$ because of noise in the timing of mitotic entry (Methods). As such, we find that the maximum p21-GFP level reached in G1$^D$ is most reflective of the time between S-phase exit in the mother cell and S-phase entry in daughter cells. Given that much weaker correlations are observed between phase lengths before and after S-phase (Methods), we suggest that stochastic events occurring in S-phase determine p21 levels in, and the length of, the combined G2$^M$ + G1$^D$ phase.

Although p21-GFP levels begin to rise during G2, p21-GFP levels further increase after mitosis (Fig. 1b,c,f). In 21% (47/219) of G1 cells, p21-GFP accumulates to a high level shortly after mitosis and these cells enter a state we define as G1pm arrest[25] (Fig. 1f; G1 length > 600 min). Cells with high p21-GFP expression have hypo-phosphorylated pRb (Supplementary Fig. 2f). Thus we propose that G1pm cells with high p21-GFP levels are blocked before RP passage.

Consistent with p21 levels being inherited by daughter cells, simple linear regression between the maximum p21-GFP level in G2$^M$ and G1pm arrest demonstrated a positive correlation ($R = 0.43^{**}$). Following both daughter cells after mitosis, in 70% (45/64) of cases both daughters cycle, in 16% (10/64) both daughter cells enter a G1pm arrest (twin arrest), while in 14% (9/64), one daughter arrests and the other cycles (single arrest). If arrest occurred by chance following mitosis, we would expect fewer than 4.4% of mitoses to result in twin arrest (21% chance of any cell arresting, $0.21^2 = 0.044$). As such, conditions leading to arrest are frequently inherited by both

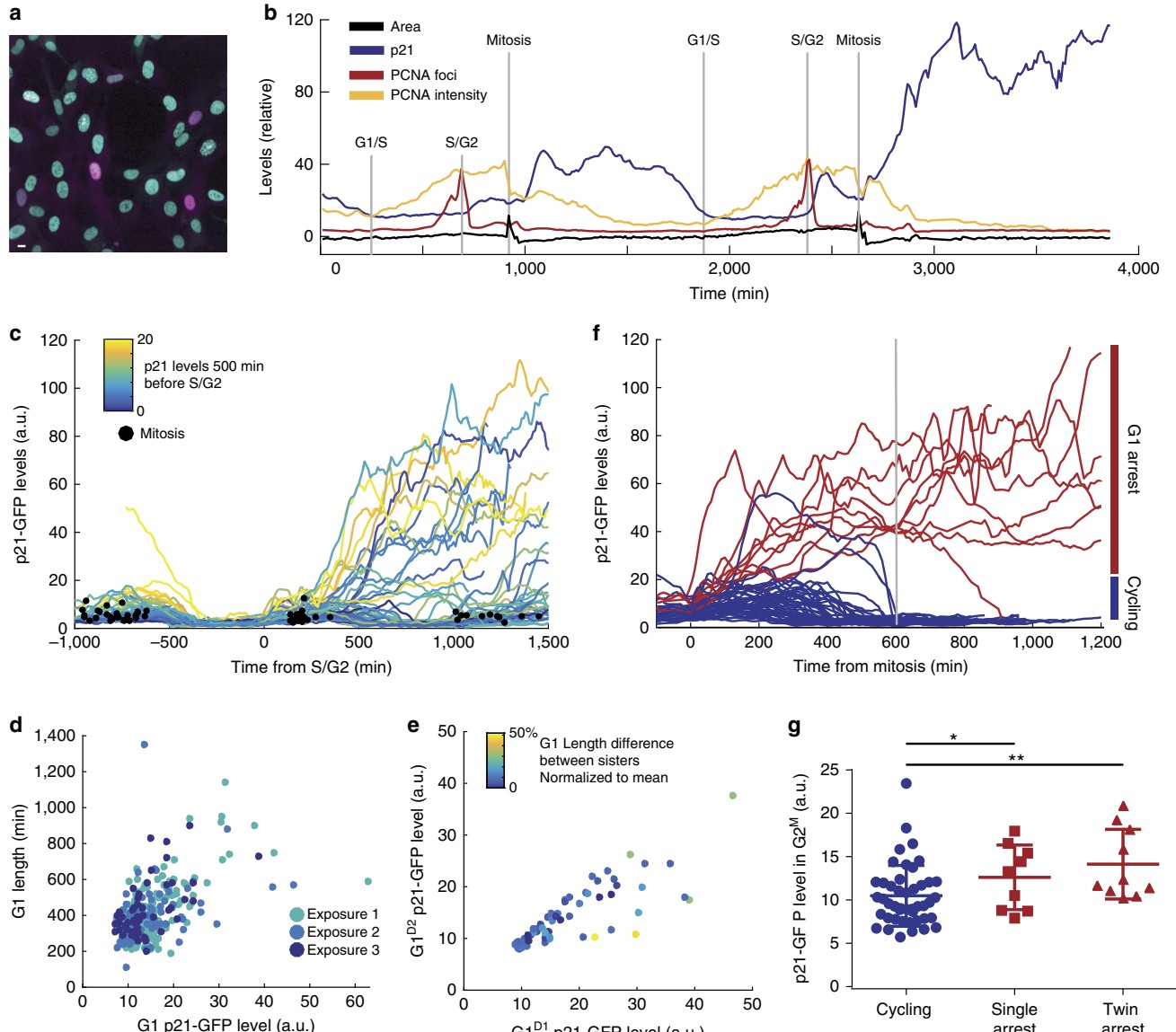

**Figure 1 | Cells exhibit cell-to-cell variability in p21 expression.** (**a**) Image of p21-GFP (magenta) and mRuby-PCNA (turquoise) expressing hTert-RPE1 cells. Scale bar is 10 μm. (**b**) Cell features captured by automated image analysis (Methods) from a single cell expressing mRuby-PCNA and p21-GFP. (**c**) Single-cell traces of p21-GFP levels aligned to S/G2; $n = 51$ cells. Tracks are coloured according to p21-GFP intensity. Black circles represent mitosis. (**d**) Correlation between maximum G1 p21-GFP intensity and G1 length (Pearson's Correlation $R = 0.62**$ ($P < 0.01$)). Three different 488 nm exposures are shown: Exp1, $n = 206$ cells; Exp2, $n = 90$; Exp3, $n = 52$ (Methods). (**e**) Correlation in G1 p21-GFP levels between sister cells—G1 Daughter1 (G1$^{D1}$) and G1 Daughter2 (G1$^{D2}$; $n = 62$, Pearson's Correlation $R = 0.81**$ ($P < 0.01$)). Data points are coloured to represent correlation between G1 length in each daughter cell, calculated as the difference between the maximum and minimum G1 length divided by the sum of the sister G1 lengths, $n = 62$. (**f**) Single-cell traces of p21-GFP levels aligned to mitotic exit; $n = 51$ cells. Grey line marks 600 min time point used to define G1pm arrest. Red curves are cells that enter a G1pm arrest, blue curves are cells that enter S-phase (cycle). (**g**) Mean p21-GFP intensity in G2$^M$ separated by daughter cell fate: cycling, $n = 49$ cells; single arrest, $n = 9$; twin arrest, $n = 10$. Significant differences are observed between arrested and cycling states using a two-sample *t*-test on log-transformed data. Error bar is s.d. *$P < 0.05$, **$P < 0.01$.

daughter cells. Finally, where either single or twin arrest occurs in daughter cells, mean G2$^M$ p21-GFP levels are higher than those in G2$^M$ of cycling daughter cells (twin arrest = 9.6 a.u.; single arrest = 8.0 a.u.; cycling = 5.9 a.u.; Fig. 1g). Thus, we propose that factors contributing to G1pm arrest are detected prior to mitosis, and the single arrest we observe is caused by asymmetric inheritance of these factors; though p21 levels can increase further in G1 to promote G1pm arrest. These observations suggest that the proliferation-quiescence decision in G1, in unperturbed cells, is regulated by p21 levels, which in turn are determined by events occurring in the mother cell S-phase.

**p53 drives p21 heterogeneity in unperturbed conditions.** We sought to determine factor(s) driving heterogeneity in p21 levels. Multiple transcription factors have been linked to p21 regulation[10]; the best characterized being p53. Depletion of p53 by siRNA resulted in a large reduction in basal p21-GFP levels, compared to control siRNA (Fig. 2a; Supplementary Fig. 3a–c). Furthermore, G1 length became more homogeneous (Fig. 2a) and no cells analysed entered G1pm arrest (Fig. 2b). Thus, both G1pm arrest and G1 delay are p53-dependent, and p53 signalling is necessary for p21 accumulation in hTert-RPE1 cells.

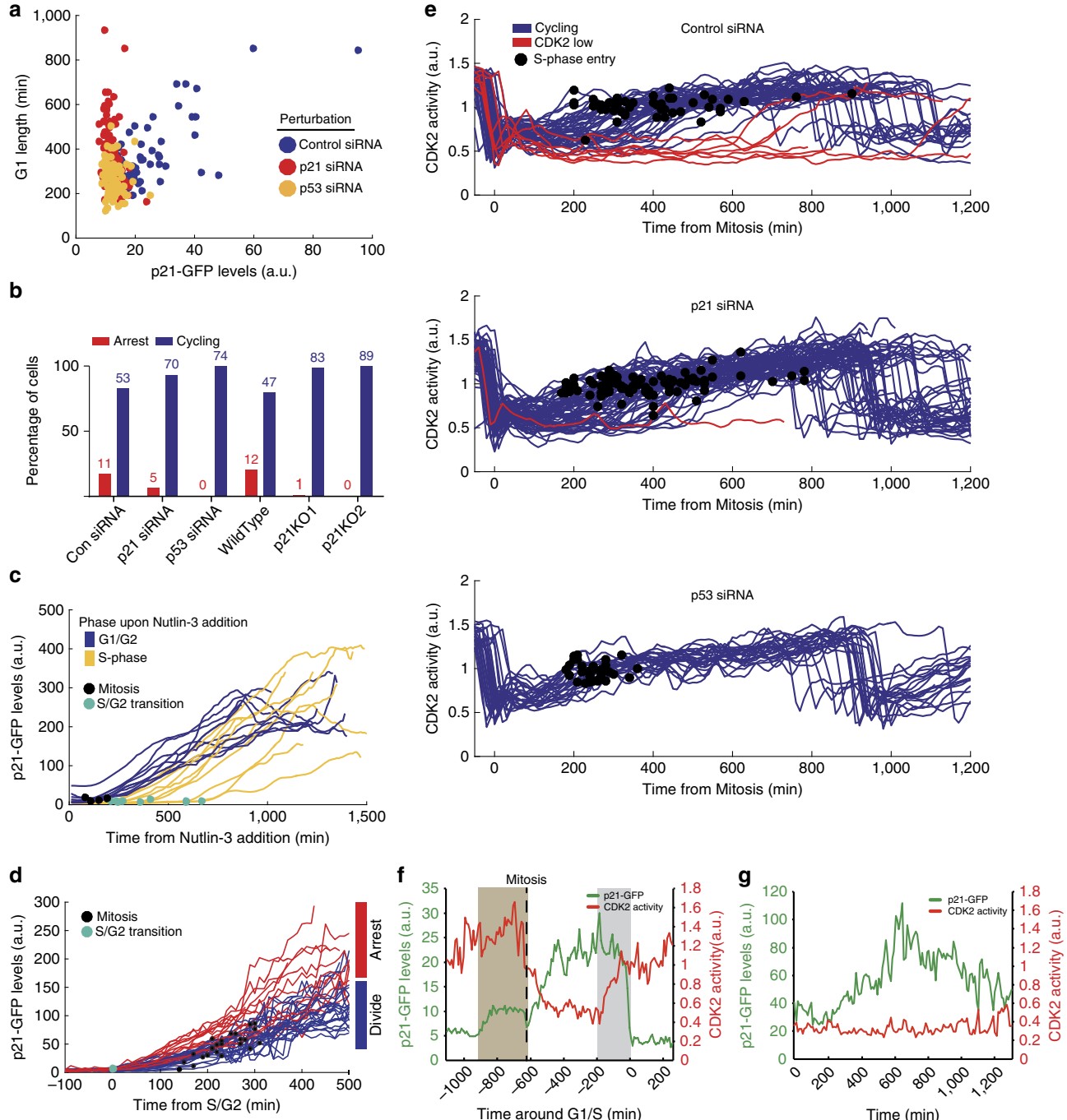

**Figure 2 | p53 drives p21 heterogeneity in unperturbed conditions.** (**a**) p21-GFP levels versus G1 length after siRNA treatment: control siRNA (blue), $n = 75$ cells; p53 siRNA (yellow), $n = 81$; or p21 siRNA (red), $n = 90$. (**b**) Percentage of cells arresting in G1 for three siRNA treatments and for p21WT versus two p21KO clones. Number of cells is shown above each bar. Red bars represent cells that enter G1pm arrest, blue bars are cells that enter S-phase (cycle). (**c**) Single-cell traces of p21-GFP levels after 5 μM Nutlin-3 addition: S-phase on addition, yellow, $n = 8$ cells; G1 on addition, blue, $n = 8$. Time of mitosis is marked by black circles and time of S-phase entry is marked by blue circles on each single-cell trace. (**d**) Effect of rate of p21-GFP accumulation on G2 entry after Nutlin-3 addition to cells in S-phase. Black circles represent mitosis. Arrest, red, $n = 19$ cells; Mitosis, blue, $n = 28$. (**e**) CDK2 activity profiles aligned to mitosis for: control siRNA, $n = 51$ cells; p21 siRNA, $n = 68$ cells; and p53 siRNA, $n = 25$ cells. Red curves represent CDK2low cells. Black circles represent time of S-phase entry. (**f**) Single-cell trace of CDK2 activity and p21-GFP levels in an unperturbed cell. Brown shading is G2. Grey shading shows a period in G1 where CDK2 activity is increasing despite expression of p21-GFP. Traces aligned to S-phase entry at $t = 0$ min. p21-GFP levels are shown in green and CDK2 activity is shown in red. (**g**) Single-cell trace of CDK2 activity and p21-GFP levels in an unperturbed cell arrested in G1. p21-GFP levels are shown in green and CDK2 activity is shown in red. Note: different scale for p21-GFP y-axis compared to (**f**).

We further analysed the dependence of p21 expression on p53 by adding the inhibitor Nutlin-3 to stabilize p53 protein[26]. On Nutlin-3 addition, we see a net increase in p21-GFP expression (Supplementary Fig. 3d). However, p21-GFP shows a cell cycle phase-dependent response to Nutlin-3. Nutlin-3-treated cells in G1 or G2 accumulate p21-GFP continuously and

arrest in G1 (Fig. 2c), while cells in S-phase maintain low p21-GFP during S-phase, before rapidly accumulating p21-GFP after the S/G2 transition (Fig. 2c; Supplementary Movies 2,3). In these cells we observe that G2 arrest correlates with the rate of p21-GFP accumulation, meaning that cells with the highest rate of p21-GFP accumulation arrest in G2, while cells accumulating p21-GFP more slowly undergo mitosis before entering a G1pm arrest (Fig. 2d; Methods).

**G1pm arrest is p21-dependent but G1 delay is p21-independent**. Since p21-GFP levels correlate with G1 length, and depletion of p53 leads to a more homogenous G1 length, we sought to determine if p21 regulates G1 length. Following p21 depletion by siRNA, only 7% (5/75) of p21-depleted cells entered a G1pm arrest, compared to 17.2% (11/64) of control siRNA-treated cells (Fig. 2b), but we saw no change in G1 length distribution (Fig. 2a). To confirm the effect of depleting p21 by siRNA, we generated CRISPR/Cas9 deletions in the *CDKN1A* gene using two different gRNAs, p21KO1 and p21KO2 (Supplementary Fig. 3e; Supplementary Note 1). We observed only one p21KO cell entering G1pm arrest (1/172; Fig. 2b), in spite of our observation that p21KO cells have increased levels of DNA damage, as measured by γH2AX levels (Supplementary Fig. 3f). As observed following p21 depletion by siRNA, p21KO in two independently generated clones did not change G1 length (Supplementary Fig. 3g). As such, we find that heterogeneity in G1 length is p53-dependent but p21-independent, while G1pm arrest is both p21- and p53-dependent. Moreover, that cells lacking p21 have increased levels of DNA damage suggests that p21 is important in maintaining long-term genome stability, consistent with recent work[27].

**High p21 levels are required to inhibit CDK2 activity**. Since p21 is a CDK inhibitor, we sought to characterize how CDK2 activity changed in p21- versus p53-depleted cells. We generated an hTert-RPE1 mRuby-PCNA cell line expressing a live cell reporter of CDK2 activity[28] and depleted p21 or p53 by siRNA. In control siRNA-depleted cells, we see heterogeneity in CDK2 activity following mitosis (Fig. 2e) and there is significant negative correlation between the minimum level of CDK2 activity and G1 length ($R = -0.67^{**}$). In addition, 13.7% (7/51) of the population enter a CDK2$^{low}$ state (red curves: CDK activity < 0.75 a.u. at 600 min, Fig. 2e), which, although arbitrarily defined, is similar to the percentage of cells entering a p21$^{high}$ state (red curves, Fig. 1f). This suggests that the CDK2$^{low}$ state corresponds to the p21$^{high}$ state and represents G1pm arrest before the RP. Indeed, if we fix unperturbed hTert-RPE1 cells co-expressing mRuby-PCNA and the CDK2 activity sensor and immunostain for p21, we see that CDK2$^{low}$ cells in G1 represent 16% of the population and that these cells express high levels of p21 (Supplementary Fig. 3h).

Consistent with p53 siRNA treatment in p21-GFP hTert-RPE1 cells, p53 depletion in CDK2 activity sensor expressing cells leads to a more homogenous G1 length and cells do not enter G1pm arrest (0/25; Fig. 2e). However, following p21 siRNA we still see a strong negative correlation between the minimum CDK2 activity and G1 length ($R = -0.631^{**}$). We only observed one p21-depleted cell entering a CDK2$^{low}$ state (1/68), suggesting that the CDK2$^{low}$ G1pm arrest state is due to p21-dependent inhibition of CDK activity, consistent with previous work[22] (Fig. 2e). To determine how different levels of p21 correlate with CDK2 activity, we imaged hTert-RPE1 p21-GFP cells co-expressing an mRuby-tagged CDK2 activity sensor. We found in cycling G1 cells that CDK2 activity is able to increase in the presence of p21 protein (Fig. 2f, grey box). Moreover,

CDK2 activity remains high throughout G2 in unperturbed cells, irrelevant of p21-GFP levels (Fig. 2f, brown box). Importantly, at high p21-GFP levels in G1 cells, CDK2 activity is inhibited and cells enter G1 arrest (Fig. 2g).

These data further support the notion that p53 signalling induces p21 accumulation over G2$^M$ and G1$^D$ and, at sufficiently high p21 levels in G1, CDK2 activity is inhibited leading to a p21-dependent G1pm arrest. However, at low-intermediate levels, p21 is a poor CDK2 inhibitor and a G1 delay is not p21-dependent.

**Basal p21 expression correlates with DNA damage foci**. Given our observation that p21 heterogeneity in unperturbed hTert-RPE1 cells is p53-mediated, reflective of the time between S/G2 in the mother and G1/S in the daughter, and that p21 starts to increase in G2, we postulated that p21 levels increase in response to endogenous DNA damage occurring during S-phase in the mother cell. To assess this, we immunostained unperturbed hTert-RPE1 cells with the DNA damage marker γH2AX and noted a striking correlation between p21-GFP expression in G1 and G2 cells and the presence of a single nuclear focus of γH2AX (Fig. 3a–c). We found a similar correlation between foci of 53BP1 and PS1981-ATM with p21-GFP levels (Supplementary Figs 4a,b). These single foci represent endogenous DNA damage (since no exogenous damaging agents were added) and resemble stretches of incompletely replicated DNA[29], leaving single-stranded DNA (ssDNA) gaps in G2 cells. Indeed, in unperturbed hTert-RPE1 cells we see colocalization of the ssDNA marker, RPA2, with single γH2AX foci in G2, and an increase in both RPA2 and γH2AX foci is seen after treatment with the DNA polymerase inhibitor, aphidicolin (Supplementary Fig. 4c). Taken together, we hypothesize that endogenous DNA damage generated in unperturbed S-phases could trigger p21 expression and be the molecular basis for the observed heterogeneity in p21 levels.

To explore the link between endogenous DNA damage and p21 expression, we generated hTert-RPE1 cells co-expressing p21-GFP and mRuby-53BP1. 53BP1 localizes to DNA Double Strand Breaks (DSBs) during G1 (ref. 30). Live cell imaging in unperturbed cells revealed that DSBs co-segregate with p21-GFP expression in G1 (Supplementary Movie 4; Supplementary Fig. 4d). To determine how the presence of DNA damage foci in G1 affects proliferation, we generated a cell line co-expressing mRuby-PCNA and GFP-53BP1. Timelapse imaging revealed that in the majority of cell divisions, sister G1 cells behave similarly such that in 47% (67/142) of mitoses neither sister has a 53BP1 focus, in 24% (34/142) both sister cells have a focus, and in 29% (41/142) only one sister has a focus (Fig. 3d). When both sister cells have a 53BP1 focus, the timing of focus appearance shows a small, but significant, positive correlation ($R = 0.39^*$; Fig. 3e). Moreover, G1 cells displaying a 53BP1 focus have a longer G1 phase than cells never displaying a focus (Fig. 3f). In 81% (48/59) of G1 cells expressing GFP-53BP1 foci, foci start to decrease in size before S-phase, suggesting that most hTert-RPE1 cells repair DNA damage prior to the G1/S transition. In unperturbed mRuby-PCNA 53BP1-GFP expressing cells, 22% (50/222) of cells enter a G1pm arrest, and of these, 82% (41/50) have at least one 53BP1 focus, indicating a high correlation between G1pm arrest and the presence of DNA damage.

These data suggest that DNA damage is inherited from the mother cell leading to 53BP1 localization to DSBs in daughter cells, that correlates with both a G1 delay and G1pm arrest. Despite p21 levels correlating with the presence of 53BP1 foci and a G1 delay, only G1pm arrest, but not G1 delay, is p21-dependent.

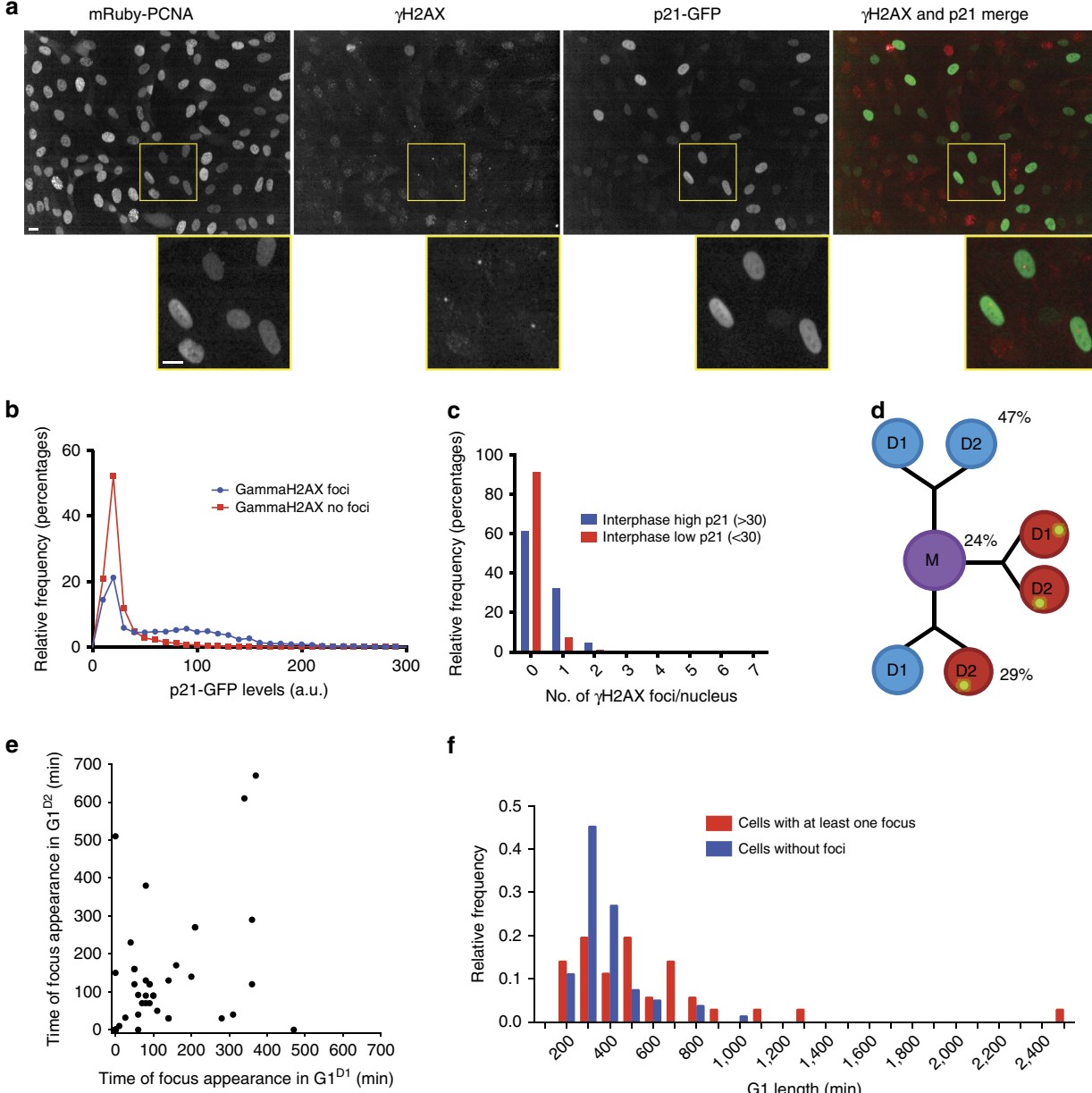

**Figure 3 | Basal p21 expression correlates with the presence of DNA damage foci.** (**a**) Images of hTert-RPE1 mRuby-PCNA p21-GFP cells fixed and stained for γH2AX. Magnified regions of each image are also shown. p21-GFP is green, γH2AX is red in merged images. Scale bars are 10 μm. (**b**) Quantification of p21-GFP nuclear fluorescence in interphase (excluding S-phase) cells with or without γH2AX nuclear foci. With foci, $n = 2086$ cells; without foci, $n = 12476$. (**c**) Quantification of number of γH2AX foci per nucleus after separation of cells based on a high or low p21-GFP nuclear fluorescence (high p21 > 30 a.u.). High p21-GFP, $n = 3896$ cells; low p21-GFP, $n = 11049$. (**d**) Fates of sister cells pairs (D1 and D2) with regards to 53BP1 focus appearance. 142 daughter pairs were analysed. (**e**) Correlation between the time of 53BP1 focus appearance in sister cells. 45 sister pairs are shown. Note: 14 sister pairs are clustered at time 0,0 min and are not visible as separate values. Pearson's Correlation $R = 0.39*$ ($P < 0.05$). (**f**) Histogram of G1 length separated by the presence or absence of a 53BP1-GFP focus. The difference between the two populations is significant (unpaired student's t-test $P = 0.0003$). 118 cells were scored manually.

**Impaired DNA replication can induce p21 expression**. Our data suggest that stochastic DNA damage occurring during normal DNA replication could generate the observed heterogeneity in p21 expression. To determine if p21 expression can be induced by impaired DNA replication during S-phase, we treated cells with low dose aphidicolin. Aphidicolin treatment caused an increase in the proportion of cells with a single γH2AX focus, in the number of γH2AX foci per nucleus, and in p53 expression (Fig. 4a; Supplementary Fig. 5a). In timelapse imaging of aphidicolin-treated cells, we observed a longer S-phase, indicative of

replication fork slowing[31], and a marked increase in p21-GFP levels following S/G2, compared to DMSO-treated cells (Fig. 4b). Following mitosis, 62.5% (20/32) of aphidicolin-treated cells entered G1pm arrest, compared to 13.2% (5/38) of DMSO-treated cells (Fig. 4b). In addition, 25% (8/32) of aphidicolin-treated cells arrested in G2 with high p21-GFP expression, a phenotype rarely observed in unperturbed cells. Of note, we observed that after aphidicolin treatment of hTert-RPE1 cells expressing mRuby-PCNA and a CDK2 activity sensor, that cells arresting in G2 eventually downregulate CDK2 activity (Fig. 4d; top panel,

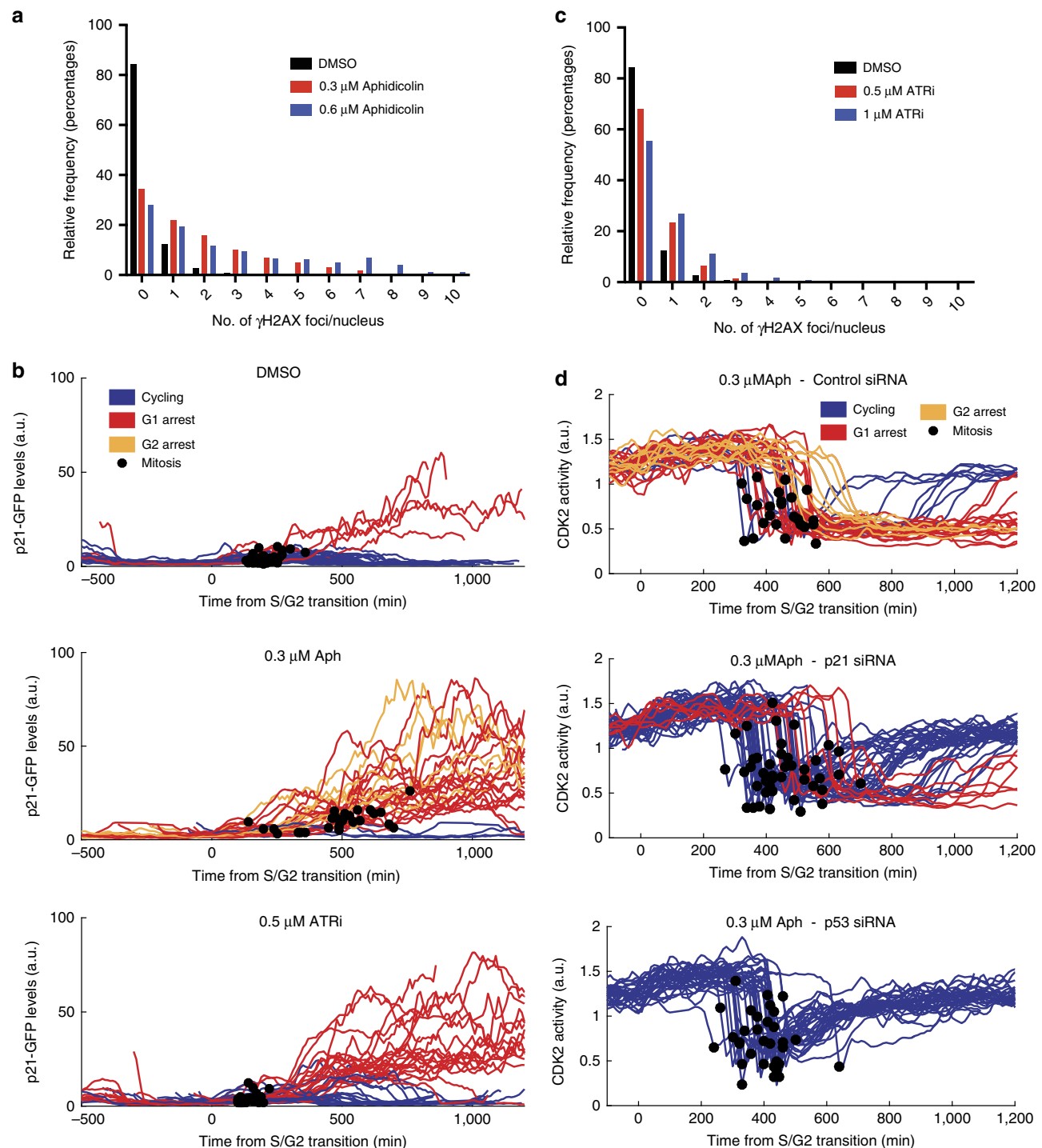

**Figure 4 | Impaired DNA replication can induce p21 expression.** (**a**) Number of γH2AX foci per nucleus in cells treated with DMSO or aphidicolin for 24 h. DMSO, n = 1527 cells; 0.3 μM Aph, n = 775; 0.6 μM Aph, n = 745. (**b**) Single-cell traces of p21-GFP expression after treatment of asynchronous cells with: DMSO, n = 38; 0.3 μM aphidicolin, n = 32; or 0.5 μM ATRi, n = 40. All traces are aligned to S-phase exit. Black circles represent mitosis. Blue curves represent cells that enter S-phase, red curves represent cells that arrest in G1 and yellow curves represent cells that arrest in G2. (**c**) Number of γH2AX foci per nucleus in cells treated with DMSO or ATRi for 24 h. n = 1527, DMSO; n = 1420, 0.5 μM ATRi; n = 1467, 1 μM ATRi. (**d**) CDK2 activity profiles in cells treated with 0.3 μM aphidicolin and: n = 34, DMSO; n = 32, p53; and n = 42, p21 siRNA, aligned to S/G2. Black circles represent mitosis. Blue curves represent cells that enter S-phase, red curves represent cells that arrest in G1 and yellow curves represent cells that arrest in G2.

yellow curves). This suggests that when levels of damaged DNA, and corresponding p21 levels, exceed a threshold, CDK2 activity is inhibited and cells undergo G2 arrest. Overall, these data show that by inducing aberrant DNA replication with aphidicolin, cells activate p21 expression, downstream of p53, which promotes either a G1 or G2 arrest.

To further examine if DNA damage during S-phase can induce p21 accumulation in G2, we treated cells with an ATR kinase inhibitor (ATRi; AZD6738) to induce destablization of stalled DNA replication forks[32]. Treatment with ATRi leads to an increase in the proportion of cells with a single γH2AX focus, in the number of γH2AX foci per nucleus, and in p53 expression

(Fig. 4c; Supplementary Fig. 5b). Similar to aphidicolin treatment, cells treated with ATRi display a marked increase in p21-GFP levels in cells in G1, compared to DMSO-treated cells (Fig. 4b). The number of cells arresting in G1 following mitosis also increased from 13.2% (5/38) in DMSO-treated cells to 52.5% (21/40) in ATRi-treated cells (Fig. 4b). However, unlike aphidicolin treatment, we saw a small, but significant, decrease in S-phase length when compared to DMSO-treated cells (Supplementary Fig. 5c), and no cells were observed to arrest in G2, consistent with G2 arrest being ATR-dependent[33]. Taken together, these data demonstrate that in the absence of ATR activity, DNA damage in S-phase leads to accumulation of p21 protein that frequently exceeds the threshold required for cell cycle arrest.

To determine if p21 and/or p53 is required for the cellular response to impaired DNA replication, we depleted p21 or p53 by siRNA and treated hTert-RPE1 cells expressing mRuby-PCNA and the CDK2 activity sensor with aphidicolin. p53-depleted cells treated with aphidicolin were unable to arrest in G1 (0/32) (Fig. 4d), G1 length was more homogenous and G2 arrest was absent. However, in a fraction of cells, an extended G2 phase was observed. In p21-depleted cells treated with aphidicolin, there was no G2 arrest (0/42; although similar to p53-depleted cells extended G2 lengths were observed), and G1pm arrest was severely compromised compared to control siRNA-treated cells (7/42 p21-depleted cells arrest versus 17/24 in control siRNA-depleted cells; Fig. 4d). These data support our conclusion that G1pm arrest is p21- and p53-dependent, and also reveal a p53- and p21-dependent G2 arrest.

**Skp2 and Cdt2 degrade p21 with different rates and timings.** Since p21-GFP is absent in all cells during S-phase, we hypothesized that, while high levels of p21 promote cell cycle arrest, its removal is required for efficient DNA replication. Both SCF[Skp2] and CRL4[Cdt2] have been implicated in p21 degradation at the G1/S transition[34–38] and we wanted to determine the relative contribution of these two pathways to p21 removal.

Following Skp2 depletion by siRNA, we find that p21-GFP is still degraded prior to and over S-phase (Fig. 5a; Supplementary Fig. 6a,b). However, p21-GFP degradation becomes switch-like at the G1/S transition in Skp2-depleted cells (Fig. 5a). While in control siRNA-depleted cells p21-GFP levels correlate with G1 length ($R = 0.76$**), in Skp2-depleted cells G1 length was longer and showed weaker correlation with p21-GFP levels ($R = 0.40$**; Fig. 5b). These data suggest that SCF[Skp2] plays an important role in degrading p21 prior to the G1/S transition and is responsible for the slower rate of p21 degradation during G1. Additionally, in control siRNA-treated cells we observed a 'burst' in p21-GFP expression after mitosis. However, following SCF[Skp2] depletion, p21-GFP levels in G2 increased much more rapidly and no post-mitotic burst in p21-GFP was observed (Fig. 5a).

After Cdt2 depletion by siRNA, notably fewer cells entered S-phase than control-depleted cells, and only cells that removed p21-GFP to undetectable levels entered S-phase (Fig. 5a,b; Supplementary Fig. 6c; Methods). In Cdt2-depleted cells, p21-GFP levels increased immediately after S-phase entry (Fig. 5a) and this increase was associated with a longer S-phase. We observed anti-correlated cycles of p21-GFP expression with mRuby-PCNA foci over short time frames—suggestive of interrupted cycles of incomplete DNA replication (Supplementary Fig. 6d; Supplementary Movie 5). Cell death was also observed following Cdt2 depletion. Finally, switch-like degradation of p21-GFP at the G1/S transition was abolished when Cdt2 was depleted (Fig. 5a). Thus, our data show that CRL4[Cdt2] is essential for degrading p21 during S-phase and is responsible for the switch-like, rapid degradation of p21 at

the G1/S transition. Cdt2 depletion phenotypes were rescued by p21 and Cdt2 co-depletion—where cells had normal S-phase lengths (Supplementary Fig. 6e), and no interrupted DNA replication or cell death was observed. These data show that even the presence of low levels of p21 during S-phase can perturb DNA replication, cause premature S-phase exit, and cell death.

Depleting both Skp2 and Cdt2 prevented the vast majority of cells from entering S-phase (Fig. 5a). Cells that did enter S-phase had p21-GFP levels below our detection limit (Fig. 5b). Of note, we observed that after depletion of only Cdt2, or after Cdt2 and Skp2 co-depletion, overall p21-GFP levels increased over time, indicative of accumulating DNA damage (Fig. 5c). Thus we find that p21 removal at the G1/S transition promotes efficient DNA replication, limiting DNA damage, and that SCF[Skp2] and CRL4[Cdt2] cooperate to degrade p21 with different rates and timings.

**Two double-negative feedback loops control p21 degradation.** To understand the mechanism behind p21 dynamics and its removal, we sought to mathematically model p21 regulation. We constructed a wiring diagram based on well-established reactions that make up the p21 control network (Fig. 6a; Methods). In our model, two double-negative feedback loops control p21 expression: (i) phosphorylation by CDK2 targets p21 for Skp2-dependent degradation[24,39], while high levels of p21 inhibit CDK2 activity[6]; and (ii) PCNA in active replication complexes (aRCs) recruits CRL4[Cdt2] to degrade p21 (refs 24,40), while p21 can bind to PCNA interfering with its function as a processivity factor during DNA replication[41,42].

Consistent with our data, stochastic simulations show p21 degradation prior to the G1/S transition, the absence of p21 during S-phase, and its re-accumulation in G2 (Fig. 6b; Supplementary Fig. 7a). p21 loss in G1 is mainly caused by increasing CDK2 activity, which promotes Skp2-dependent degradation at a comparatively low rate. Once CDK2:Cyclins reach a threshold, PCNA loading onto DNA is triggered, aRCs accumulate, and strong Cdt2-dependent degradation removes residual p21 while the cell enters S-phase. At the end of S-phase, aRCs disappear and p21 re-accumulates. We also see significant intra- and inter-cellular variability in p21 levels due to the stochasticity of DNA damage and p53, supporting our experimental observations (Fig. 6c). However, p21 remains low over S-phase even when damage occurs, consistent with data from our experiments (Figs 2c,d and 4b). This is due to the presence of a bistable switch, which creates two distinct cellular states with either high or low p21 level depending on the cell cycle stage (Fig. 6d). In G1, CDK2 activity is low and RCs are inactive such that p21 is attracted to the upper steady state. As CDK2:Cyclin accumulates and origins fire, p21 is lost and cells enter S-phase. The existence of a bistable region between both states allows the S-phase level of p21 to be insensitive to DNA damage, while damage in G1 (and G2, Supplementary Fig. 7b) leads to an increase in p21 and in the CDK2:Cyclin threshold required to enter S-phase. Hence, large amounts of damage can cause G1pm arrest (Fig. 6d, topmost curve).

We sought to test the relative importance of the Skp2- and Cdt2-degradation pathways for bistability. Skp2 depletion leaves the bistable switch intact such that S-phase levels of p21 remain low (Supplementary Fig. 7c). However, Skp2-depleted cells show a more rapid p21 degradation prior to S-phase and a more rapid re-accumulation in G2 (Fig. 6e, left panels). Cdt2 depletion almost completely abolishes bistability, which makes the G1/S transition easier to reverse (Supplementary Fig. 7c), and leads to p21 expression during S-phase (Fig. 6e, right panels). This latter phenotype is more pronounced in cells depleted of

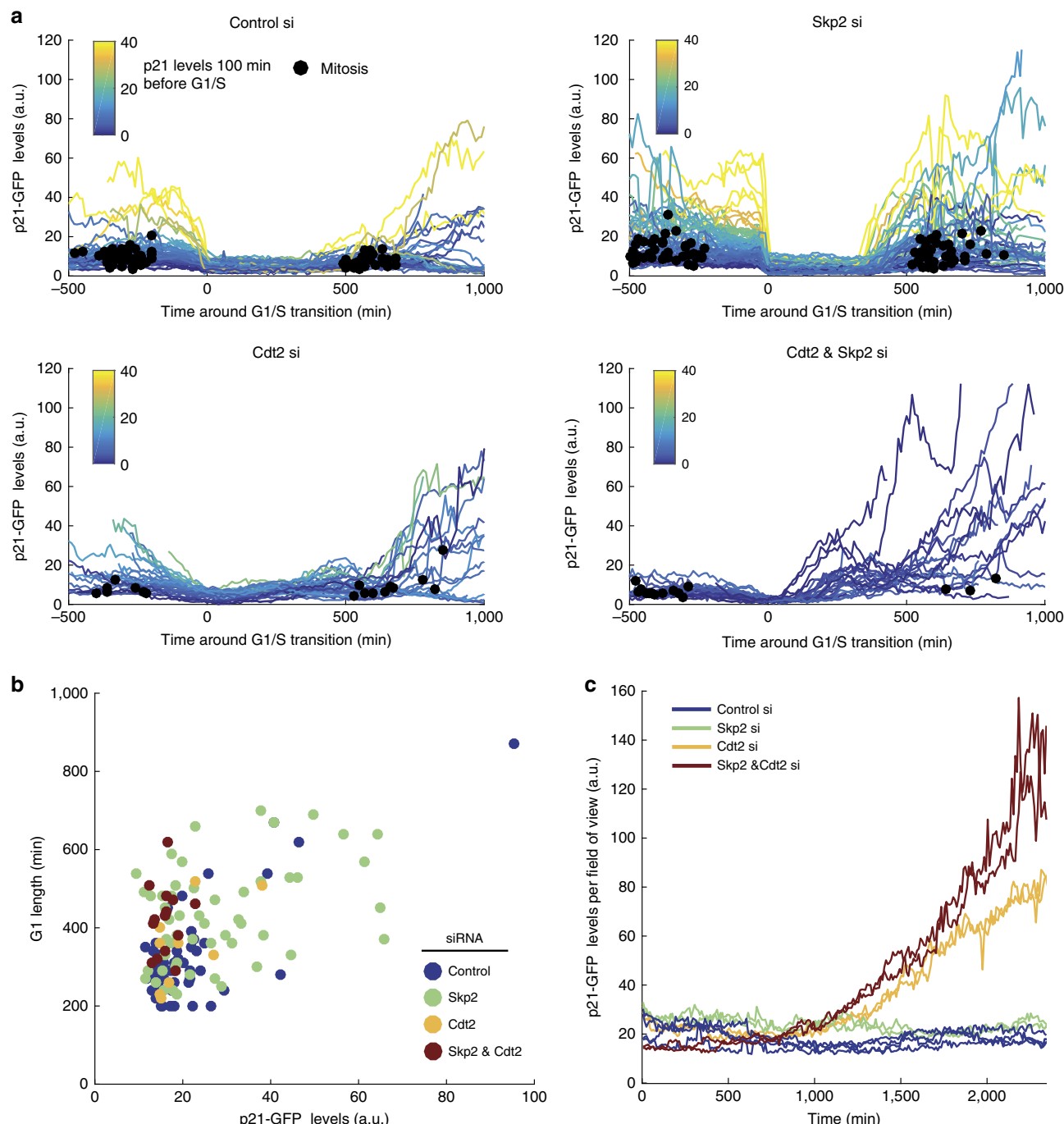

**Figure 5 | Skp2- and Cdt2-dependent pathways degrade p21 with different rates and timings.** (**a**) Single-cell traces, coloured by p21-GFP intensity in G1, aligned to the G1/S transition. Four siRNA treatments are shown: $n = 65$ cells, control; $n = 81$, Skp2; $n = 37$, Cdt2; $n = 25$, Skp2 and Cdt2 co-depletion. Black circles represent mitosis. Note: only cells entering S-phase are shown. (**b**) Graph shows correlation between p21-GFP intensity and G1 length for each siRNA condition shown in (**a**): $n = 50$ cells, control; $n = 54$, Skp2; $n = 9$, Cdt2; $n = 14$, Skp2 and Cdt2 co-depletion. (**c**) Graph shows average p21-GFP nuclear level across FOV for four different siRNA treatments. Four FOVs are shown for control siRNA cells and two FOVs are shown for the other treatments, each curve is one FOV. Control siRNA is shown in blue, Skp2 siRNA is shown in green, Cdt2 siRNA is shown in yellow and double Skp2 and Cdt2 depletion is shown in red.

both Skp2 and Cdt2 (Supplementary Fig. 7d). As found experimentally (Supplementary Fig. 6d,e), p21 expression during S-phase can cause premature S-phase exit, which modelling suggests is due to PCNA inhibition (Supplementary Fig. 7e). Hence, strong Cdt2-dependent degradation of p21 ensures an irreversible G1/S transition and undisturbed DNA replication.

In summary, we have developed a model that not only explains p21 dynamics in unperturbed and DNA damaged cells but also correctly predicts the effects of Skp2 and Cdt2 depletion experiments. Moreover, modelling confirms that stochastic DNA damage events are a major source of inter- and intra-cellular heterogeneity in p21 levels, and reveals a

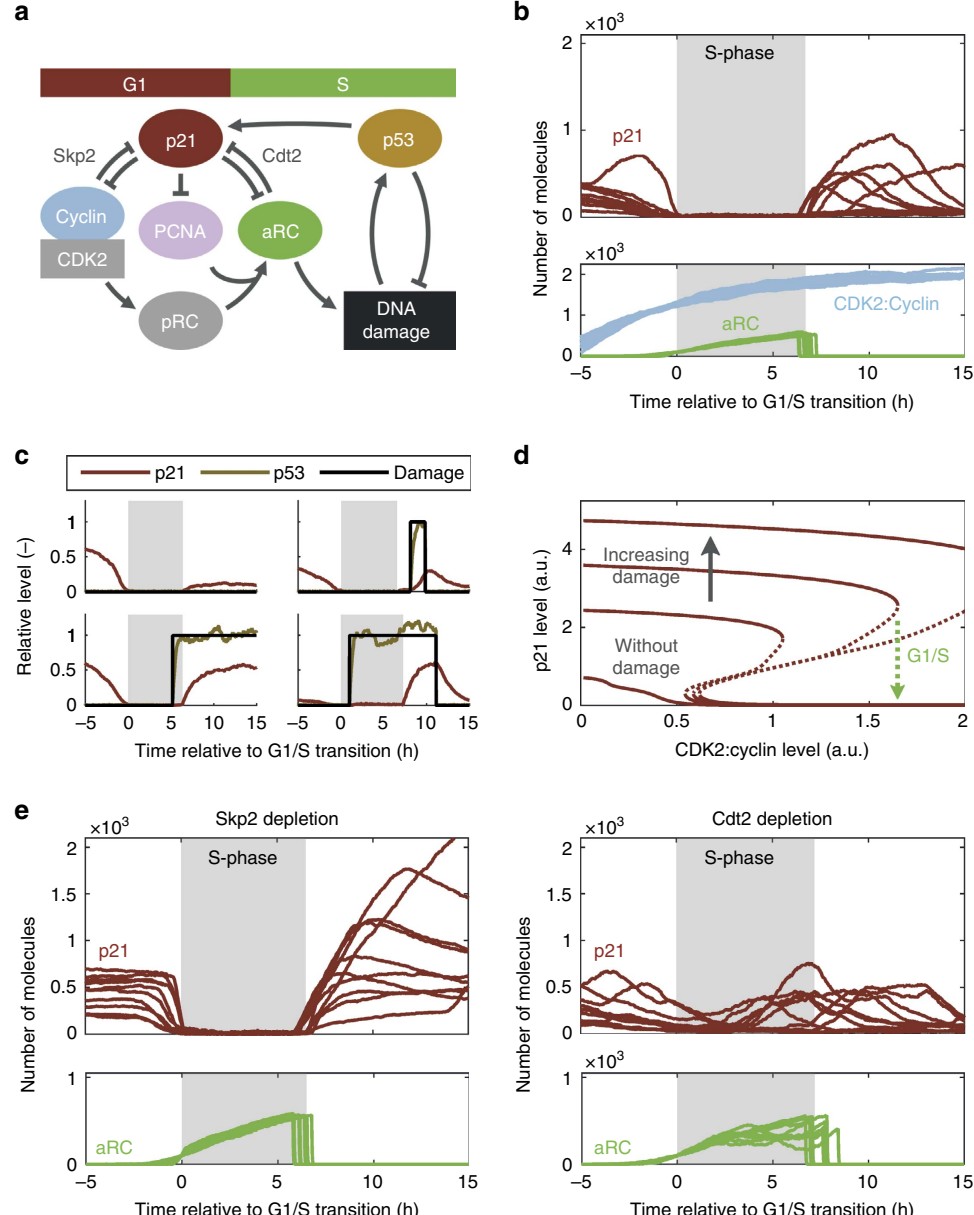

**Figure 6 | Two double-negative feedback loops control p21 degradation.** (**a**) Influence diagram of the p21 regulatory network. p21 is engaged in two mutually inhibitory motifs with CDK2:Cyclin and aRCs. CDK2 activity primes replication complexes (pRC) for PCNA loading, which facilitates S-phase entry and DNA synthesis by aRCs. The latter can cause DNA damage and p53 expression, promoting p21 synthesis and DNA repair. (**b**) Stochastic simulation of p21 (top), and total CDK2:Cyclin and aRCs (bottom), relative to the G1/S transition; $n = 10$. Grey shaded regions indicate S-phase. (**c**) Relative level of p21, p53 and DNA damage in four simulated cells. Grey shaded regions indicate S-phase. Note p21 absence in S-phase even in cells that suffer damage (for example, lower right). (**d**) Stable (solid) and unstable (dashed) steady states of p21 with respect to the total level of CDK2:Cyclin for increasing DNA damage. The G1/S transition is indicated for intermediate levels of damage. (**e**) Stochastic simulation of p21 (upper panels) and aRCs (lower panels) in Skp2-depleted (left) and Cdt2-depleted (right) cells; $n = 10$.

bistable switch that makes S-phase cells unable to accumulate p21 protein.

**Arrest in response to serum starvation is not p21-dependent.** Given that passage through the RP depends on mitogen-signalling and thus serum concentration, we wanted to determine how p21-GFP levels change in hTert-RPE1 cells during G1pm arrest in response to serum starvation. As cells exit mitosis in the absence of serum, the majority (90%; 26/29) enter G1pm arrest (Fig. 7a). Importantly, these arresting cells have variable p21-GFP levels, and many serum-starved G1pm-arrested cells have low p21-GFP levels, comparable to those measured during

G1 in cycling cells (Fig. 1f). Moreover, unlike cells in serum, $G2^M$ p21-GFP levels do not correlate with G1pm arrest ($R = 0.0989$, $P = 0.6098$). Therefore, while G1pm arrest in the presence of high serum is characterized by high p21 expression, G1pm arrest after serum withdrawal is not.

Since we did not observe a correlation between p21-GFP levels and G1pm arrest after serum starvation, we wanted to know if cells could arrest in G1 in response to serum withdrawal in the absence of p21 protein. We serum-starved WT, p21KO1 and p21KO2 hTert-RPE1 cells and measured their cell cycle profiles by FACS. We found that p21KO cells arrested in G1 as efficiently as WT cells in response to serum withdrawal (Fig. 7b).

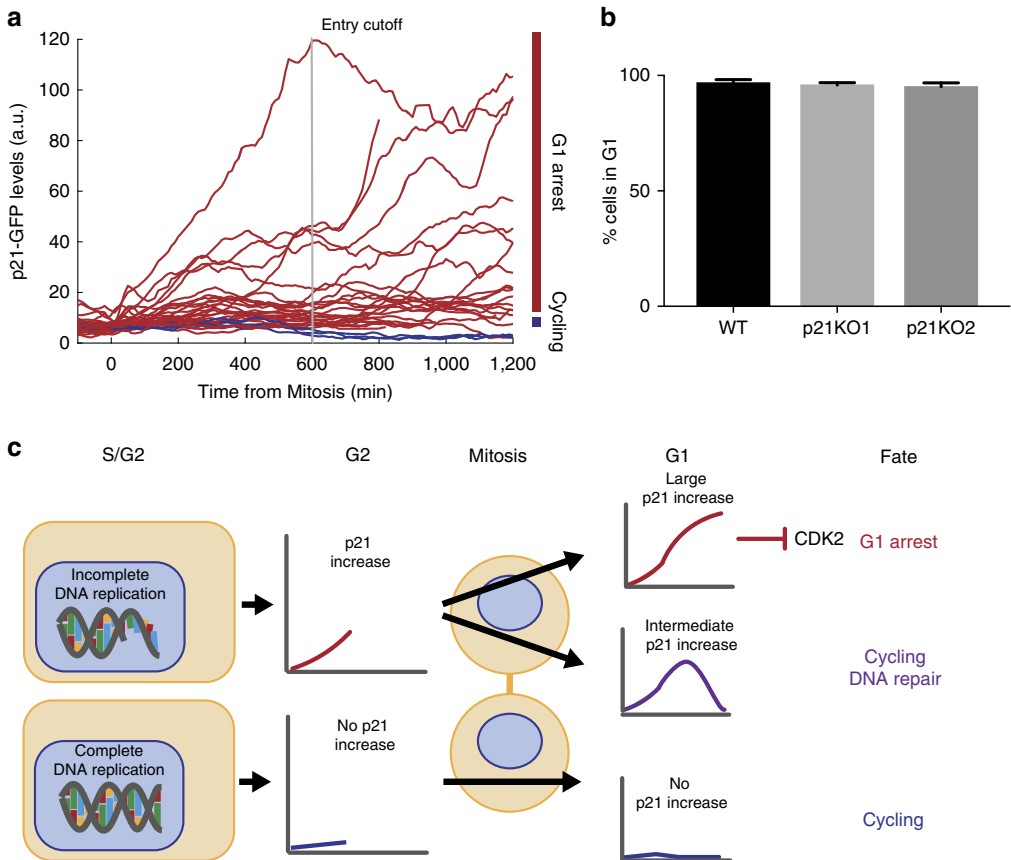

**Figure 7 | Arrest in response to serum starvation is not p21-dependent.** (**a**) Single-cell traces of p21-GFP levels aligned to mitotic exit (serum was withdrawn in the previous cycle); $n = 29$ cells. Grey line marks 600 min time point used to define G1pm arrest. Red curves represent cells entering G1pm arrest, blue curves represent cells that enter S-phase. (**b**) Graph showing the percentage of cells arresting in G1 in three different mRuby-PCNA hTert-RPE1 cell lines (WT and two p21KO clones) after serum withdrawal for 24 h. Mean and s.d. of two independent FACS experiments are shown. (**c**) Model for how p21 propagates information on DNA damage in the mother cell to influence the proliferation-quiescence decision in daughter cells. At intermediate levels of p21 (purple curve) cells still cycle, albeit with a G1 delay that is p53-dependent but p21-independent.

This indicates that p21 is not required for G1pm arrest after serum removal.

Thus there exist two molecularly distinct G1pm arrest states—one that occurs in the presence of serum and is p21-dependent, and another that occurs following serum withdrawal that is p21 independent. Furthermore, the p53- and p21-dependent G1pm arrest we observe in cells grown in high serum is principally caused by endogenous DNA damage.

## Discussion

In this study, we sought to characterize and understand the role of p21 in regulating the proliferation-quiescence decision and genome stability in unperturbed cells. In response to DNA damage generated during S-phase, p21 accumulation begins in G2$^M$ and continues into G1$^D$, where, at sufficiently high levels, p21 inhibits CDK activity and promotes G1 arrest following mitosis.

Our experiments demonstrate that in unperturbed cycling cells, heterogeneity in p21 levels is fully accounted for by p53. This is consistent with a recent report demonstrating that, in cells expressing both p53-YFP and p21-CFP, p21-CFP levels peak following peaks in p53-YFP (ref. 43). The authors also observed a spike in p53 and p21 levels following mitosis, agreeing with our observation that p21-GFP levels increase rapidly in G1 post-mitosis. This may be explained in part by ssDNA breaks generated by incomplete DNA replication in the mother cell

being converted to DSBs during mitosis[44,45]. A post-mitotic increase in p21 levels can also be explained by SCF$^{Skp2}$-mediated p21 degradation being turned off in G1 by increasing APC/C$^{Cdh1}$ activity[46,47]. Since we observe significantly higher levels of p21 in G2, as well as a reduced p21 increase in G1, following Skp2 depletion, this suggests that the post-mitotic burst in p21 is likely to be a combination of both SCF$^{Skp2}$-inactivation and additional DNA damage occurring over mitosis.

Spencer *et al.*[22] demonstrated that p21 is responsible for a bifurcation in CDK2 activity following mitosis, and that this underpins the proliferation-quiescence decision in cycling cells. We find that the source of bifurcation in the p21-dependent proliferation-quiescence decision is endogenous DNA damage occurring in the mother cell (Fig. 7c). Strikingly, we find that while p21 is responsible for G1pm arrest, heterogeneity in G1 length is not p21-dependent. Because heterogeneity in G1 length is p53-dependent, this explains the correlation between p21 levels and G1 phase length; p53 activity is leading to transcriptional activation of p21, as well as causing a G1 delay. Thus, p21 is acting as a readout of p53 activity, while having an insignificant effect on G1 phase length. Determining the mechanisms responsible for this p53-dependent G1 delay remains an open challenge.

Our data and modelling suggest that p21 degradation has the hallmarks of bistability due to two double-negative feedback loops involving SCF$^{Skp2}$ and CRL4$^{Cdt2}$. Thus we provide experimental

evidence for, and extend on, previous work which concluded that the decision to remain quiescent or proliferate is bistable because of feedback regulation between p21 and CDK2 via SCF$^{Skp2}$ (ref. 21). Moreover, our finding that CRL4$^{Cdt2}$ is required for p21 degradation at the G1/S transition and over S-phase is consistent with previous work by Kim *et al.* who find, in line with our experiments, that DNA re-replication occurs in the absence of CRL4$^{Cdt2}$ and that this is p21-dependent[38].

We speculate that p21 acts downstream of pulsatile p53 signalling to integrate information on DNA damage during mother G2/M- and the daughter G1-phases that then determines the outcome of the proliferation-quiescence decision in G1. Complete degradation of p21 before S-phase entry would ensure that information on DNA damage generated during the previous cycle is fully erased before cells commit to a new cell cycle. This would allow the process of gathering information on DNA damage to begin anew following S/G2 passage. The presence of an information-gathering phase and a reset phase would allow cells to make a decision on whether to proliferate or quiesce in response to DNA damage that remains consistent across populations and generations of cells.

## Methods

**Cell culture.** hTert-RPE1 cells were from ATCC and were maintained in DMEM (Gibco) + 10% FBS (Sigma) and 1% Penicillin-Streptomycin (P/S; Gibco) at 37 °C and 5% $CO_2$. Cell lines were checked by PCR on a monthly basis for absence of mycoplasma.

**Generating mRuby-PCNA tagged hTert-RPE1 cell line.** One allele of the endogenous *PCNA* gene was tagged at the N-terminus with a cDNA encoding the mRuby fluorophore using AAV-directed homologous recombination. To generate the targeting construct, Left and Right Homology Arms (LHA, RHA) flanking mRuby cDNA were amplified by PCR from hTert-RPE1 genomic DNA using the following primers: P1 NotI: 5′–cggccgcgaatcttcagttccctcaacaa-3′ and P2: 5′-ctttgatcaggctgtttgccatggtggcggagtggcaacaac-3′ to amplify the LHA including the first 20 bp of mRuby; P3: 5′-gttgttgccactccgccaccatggcaaacag cctgatcaaag-3′ including the first 20 bp of PCNA LHA; and P4 XbaI: 5′-ctcgaat ctagatcctcctcccctccgcccaggccggc-3′ to amplify mRuby and create a Gly-Gly-Gly-Ser-Arg linker in front of the first PCNA exon. P5 XbaI: 5′-tctagattcgaggcg cgcctggtc-3′ and P6 NotI: 5′-gcggccgcacggaggaggatcacttgagg-3′ to generate the RHA.

LHA and mRuby fragments were fused by PCR using P1 and P4 and cloned by three-way ligation, together with the RHA into the NotI sites of pAAV-MCS (Stratagene). The complete targeting construct was packed into pAAV viruses using an AAV helper-free system (Stratagene), according to the manufacturer's instructions, followed by transduction of hTert-RPE1 cells. Four days after transduction, single mRuby positive integrands were sorted by flow cytometry into 96-well plates and analysed for correct targeting of PCNA by Western blotting.

**Generating p21-GFP-tagged hTert-RPE1 cell lines.** The endogenous *CDKN1A* gene was tagged at the C-terminus using CRISPR-mediated gene tagging. The gRNA 5′-TTTGAGGCCCTCGCGCTTCCAGG-3′ (PAM site in bold) downstream of the *CDKN1A* stop codon was selected using crispr.mit.edu. Forward and reverse oligos for the gRNA were annealed and ligated into the BbsI cut pX330 bicistronic Cas9 and sgRNA expression plasmid[48]. pX330-U6-Chimeric_BB-CBh-hSpCas9 was a gift from Feng Zhang (Addgene plasmid # 42230).

For the homology donor plasmid, we PCR amplified *CDKN1A* homology arms from hTert-RPE1 genomic DNA using the high-fidelity Q5 DNA polymerase (NEB). To PCR amplify the LHA we used the forward primer with a NotI site: 5′-gcggccgccaggcccttgccaagct-3′ and the reverse primer with a SalI site: 5′-gtcgacgggcttcctcttggagaa-3′. To PCR amplify the RHA we used the forward primer with a SpeI site 5′- actagttccgcccacaggaagcctgcagttctgg-3′ (underlined base represents mutated base to change the PAM site in the recombined gene) and the reverse primer with a NotI site: 5′-gcggccgc ataccctaacacagagataaccccact-3′. GFP cDNA was PCR amplified from peGFP-C1 vector (Clontech) with the forward primer with a SalI site and a poly-glycine linker: 5′-gtcgacggaggaggagtgagcaagggcgaggag-3′ and the reverse primer with a SpeI site: 5′-actagtttacttgtacagctcgtc-3′. 2LHA, RHA and eGFP PCR products were ligated into the blunt-end cloning vector pJet1.2 (Fermentas) for restriction digest. LHA, RHA and eGFP digested inserts were ligated into NotI cut pAAV-MCS vector by 4-way ligation at a ratio of vector:inserts of 1:2:2:2, using T4 DNA ligase (NEB). All constructs were checked by sequencing before transfection into cells.

To generate endogenously tagged p21-GFP cells, the pX330 p21 gRNA plasmid and the p21 homology donor plasmid were transfected into hTert-RPE1 cells

at a ratio of 1:1 using Lipofectamine 2000, according to the manufacturer's instructions (Invitrogen). Cells were allowed to expand for 7 days before FACS sorting to select GFP positive cells. Individual GFP expressing cells were sorted into 96-well plates containing 50:50 conditioned:fresh growth media. Clones were left to expand for 10–14 days and then visualized on the Opera high-throughput microscope (PerkinElmer) to determine which clones had GFP expression in the nucleus. Candidate clones were expanded and further validated, as described in Supplementary Fig. 1.

**Western blotting.** Samples were directly lysed by addition of Laemmli buffer. Whole cell lysates were loaded onto 4–20% Tris-Glycine Novex precast gels (ThermoFisher) followed by transfer to PVDF membranes. After protein transfer, membranes were briefly washed in MeOH and then dried between sheets of filter paper. When completely dry, primary antibody diluted in 5% milk in TBS was added and membranes were incubated overnight at 4 °C with rocking. Membranes were washed three times in TBS + 0.05%TritonX-100 (TBS/T) and anti-mouse or anti-rabbit HRP-conjugated secondary antibodies (CST) were diluted 1:2,000 in 5% milk in TBS and incubated with membranes for 1 h at RT, with rocking. Membranes were washed three times in TBS/T and visualized using Pierce ECL Western Blotting Substrate (ThermoScientific). All blots were scanned on the Azure c300 imaging system.

Antibodies used in this study for Western blotting are p21 (BD, ms, 556430, 1:1,000), Skp2 (Proteintech 15010-1-AP, 1:500), Cdt2 (Bethyl A300-948A, 1:250), p53 (CST 2527, 1:2,000) and GAPDH (Novus NB300-221, 1:5,000).

**Coimmunoprecipitation.** To determine if p21-GFP could bind to CDK2, we performed immunoprecipitation of GFP-tagged p21 using the μMACS GFP Isolation kit (Miltenyi). Immunoprecipitations were performed simultaneously from asynchronous mRuby-PCNA hTert-RPE1 cells and p21-GFP mRuby-PCNA hTert-RPE1 cells, according to the manufacturer's instructions. mRuby-PCNA hTert-RPE1 cells served as a negative control since they do not express GFP. Eluted samples were separated by SDS-PAGE, transferred to PVDF and probed with anti-p21 and anti-CDK2 antibodies.

**Growth curves.** Cells were plated at a density of 50,000 cells per well in 6-well plates. Duplicate cell counts were taken every 24 h using the Countess cell counter (Invitrogen).

**Generating p21KO cell lines.** We generated two CDKN1A nickase targeting vectors, based on the methods described in ref. 49. For p21KO1, we used the two gRNA sequences: 5′-CCATTAGCGGCATCACAGTCG-3′ and 5′-TCCAGGAGGC CCGTGAGCGA-3′. For p21 KO2, we used the two gRNA sequences: 5′-TCAGCT GCTCGCTGTCCACT-3′ and 5′-CCGCGACTGTGATGCGCTAA-3′. These were annealed and ligated into the dual sgRNA All-in-One Cas9$^{D10A}$ GFP nickase vector[49]. After sequencing, plasmids were transfected into hTert-RPE1 cells expressing mRuby-PCNA. Three days post transfection, GFP positive cells were sorted to single-cell density into three 96-well plates containing 1:1 conditioned:fresh growth media. Clones were left to expand for 7–10 days and then screened for p21 expression by immunostaining. Clones deemed to be negative based on p21 immunostaining were characterized further by genotyping.

For genotyping of p21KO lines, genomic DNA was extracted using the Flexigene DNA kit, according to the manufacturer's instructions. The p21 gene was amplified using nested PCR and primers: CDKN1A_KO_FWD1 5′-ATGTCAG AACCGGCT-3′ and CDKN1A_KO_REV1 5′-TTAGGGCTTCCTCTT-3′, followed by CDKN1A_KO_FWD2 5′-GGATGTCCGTCAGAACCCAT and CDKN1A_KO_REV2 5′-GTGGGAAGGTAGAGCTTGGG-3′. The resultant PCR product was ligated into the pJet1.2/blunt Cloning Vector (Fermentas). Plasmid DNA was purified from individual colonies using the Qiagen Miniprep kit and sent for sequencing using the pJet_forward primer. The results of the genotyping are shown in Supplementary Note 1. Note that for p21KO1A, 10/10 colonies showed the same genotype suggesting that both alleles have the same deletion.

**Live cell imaging.** Live cell imaging was performed on the Opera HC spinning disk confocal microscope (PerkinElmer), with atmospheric control to maintain cells at 37 °C, 5% $CO_2$ and 80% humidity. Cells were plated on 384 well CellCarrier (PerkinElmer) plates at a density of 1,000 cells/well 24 h prior to imaging. Cells were imaged using a 20X (N.A. 0.45) objective at 10 min intervals for 24–72 h in Phenol-Red Free DMEM + 10% FBS and 1% P/S. Image processing was performed in Matlab using custom-made scripts.

**Testing 488 nm exposures for live imaging.** Since laser irradiation can induce DNA damage, and thus p21 expression, we wanted to be sure that our imaging conditions were not inducing p21 expression. Therefore, we imaged hTert-RPE1 mRuby-PCNA p21-GFP expressing cells using two 488 nm laser exposures where the laser power was greater than the 20% used in our other experiments; exposure time remained constant (120 ms). The results are shown in Fig. 1d. Exposure

1 = 20%, Exposure 2 = 30%, Exposure 3 = 40% of 488 nm laser power. Across these three different exposures, we saw no difference in cell cycle timing or amount of p21-GFP expressed. Moreover, a two-sample Kolmogorov–Smirnov test produced the null result indicating both samples came from the same continuous distribution for both maximum G1 p21 after correction for 1.5X and 2X exposure difference: 20% versus 30% $P = 0.938$ and 20% versus 40% $P = 0.243$; and G1 length with no correction: 20% versus 30% $P = 0.916$ and 20% versus 40% $P = 0.607$.

**Inhibitors.** Inhibitors used were ATRi (AZD6738; Selleckchem), Aphidicolin (Sigma), Camptothecin (Sigma), Etoposide (Sigma) and Nutlin-3 (Sigma). All drugs were reconstituted in DMSO and stored in aliquots at $-20\,^\circ$C.

**siRNA transfection.** For live cell imaging, cells were either transfected 24 h prior to imaging, or, in cases where we wanted to capture the first cell cycle after protein loss, 6 h prior to imaging. Cells were transfected with 20 nM final concentration of siRNA using Lipofectamine RNAiMax, according to the manufacturer's protocol (Invitrogen). Briefly, per well of a 384 well plate, 40 nl of Lipofectamine RNAiMAX was mixed with siRNA in 10 μl OPTIMEM. To this, 20 μl cells in media, at a density of $5 \times 10^4$ cells ml$^{-1}$ was added, and cells were incubated at 37 °C. siRNAs used in this study were OnTARGETplus pools (Dharmacon).

**Cloning 53BP1-mRuby.** GFP-53BP1 plasmid was a generous gift from Jiri Lukas[50]. This plasmid contains full-length mouse 53BP1 cDNA tagged with GFP. To visualize endogenously tagged p21-GFP and 53BP1 in the same cell line, we changed the GFP tag for an mRuby tag in the 53BP1 plasmid. We removed the GFP tag with BmtI and BspEI double digestion. We PCR amplified mRuby with following primers: forward 5′-gctagcatgaacagcctgatcaa-3′ and reverse 5′-tccggaccctccgcccaggccggcga-3′. Digested mRuby PCR product was ligated into the cut 53BP1 vector and sequenced to check for in-frame ligation.

**Generating GFP- and mRuby-53BP1 stable cell lines.** hTert-RPE1 cells were plated in six well plates to ensure a density of 80% confluency on the day of transfection. Cells were transfected with 4 g of plasmid using Lipofectamine 2000, according to the manufacturer's instructions (Invitrogen). After 24 h incubation, cells were transferred to a T-25, and after a further 24 h, cells were sorted to single-cell density in 96-well plates containing 50% fresh media:50% conditioned media with 0.25 mg ml$^{-1}$ G418 (Sigma), using the BD FACS Aria. After 10–14 days selection, single-cell clones were visualized for appropriate expression of 53BP1 and those clones were expanded.

**Immunostaining.** Cells were fixed in an equal volume of media to warm 8% formaldehyde in PBS for 5 min at 37 °C. Cells were permeabilized in PBS/0.2% TritonX-100 for 5 min at RT and then blocked in 2% BSA in PBS (Blocking buffer) for 1 h at RT. Cells were incubated in primary antibody diluted in Blocking buffer for either 2 h at RT or overnight at 4 °C. Cells were washed three times in PBS and then incubated with Alexa-conjugated secondary antibodies (ThermoFisher) diluted in Blocking Buffer for 1 h at RT. Cells were washed three times in PBS and incubated in 1 μg ml$^{-1}$ Hoechst 33258 (Sigma) diluted in PBS for 15 min at RT.

Cells were imaged on the Opera HC spinning disk confocal microscope (PerkinElmer). All imaging data was uploaded to the Columbus image analysis database (PerkinElmer) for visualization and analysis.

For RPA2 immunostaining, cells were plated onto glass coverslips. Cells were briefly washed in PBS and then pre-extracted using ice-cold PBS/0.2% TritonX-100 for 1 min. Cells were then fixed in warm 4% formaldehyde and processed as usual. Cells on coverslips were imaged on a Zeiss LSM710 confocal microscope using a 60x oil, N.A. 1.4 objective.

Primary antibodies used for immunostaining in this study are: p21 (BD 556430, 1:1,000), p53 (CST 2527, 1:1,000), 53BP1 (CST 4927, 1:1,000), γH2AX (CST 9718, 1:2,000), P-S1981-ATM (CST 4526, 1:500), RPA2 (Abcam ab2175, 1:250).

**Automated image analysis of fixed cells.** All quantitative image analysis on fixed cells was performed using Columbus software (PerkinElmer). Individual protocols are described below.

*Measuring nuclear intensities of proteins.* Nuclei were segmented based on Hoescht intensity. Nuclei at the edge of the image and nuclei <100 μm$^2$ were excluded. Mitotic nuclei were also excluded by excluding nuclei with Hoescht intensity above a defined threshold (that was defined individually for each experiment based on Hoescht staining intensity and imaging conditions). The fluorescence intensity of individual proteins was then calculated in each interphase nucleus.

*Measuring cytoplasmic intensities of proteins.* Nuclei were segmented as above. To measure cytoplasmic intensities of proteins, we defined a cytoplasmic nuclear ring region of four pixels beyond the nucleus. This was to exclude any effect cell shape or area may have on cytoplasmic intensity readings. The cytoplasmic fluorescence intensity of individual proteins was then calculated in each ring region.

*Counting nuclear foci.* Nuclei were defined as above. To define and count γH2AX foci within nuclei, we needed to distinguish between the bright, individual

foci observed in G1 and G2 cells, from the dim, multiple γH2AX foci observed in most S-phase cells. We segmented foci using the 'Find Spots' function in Columbus. Specifically, we used Method A and defined the Relative Spot Intensity as 0.290 and the Splitting Coefficient as 1. We validated our automated focus finding against manual scoring of foci.

*Excluding S-phase cells.* When scoring p21-GFP levels and foci in cells, we wanted to exclude S-phase cells from the analysis since p21 levels are negligible in all S-phase cells. To exclude S-phase cells, we used a Linear Classifier in Columbus. Inputs to the Linear Classifier were a combination of Nuclear Morphology (Area, Roundness, Width, Length, Ratio Width to Length), Hoescht Texture Properties (SER Features), PCNA Intensity Properties, and PCNA Texture Properties (SER Features).

**Automated cell tracking.** Live imaging data was analysed using custom image analysis scripts (see below and Supplementary Software 1). The exception to this was analysis of hTert-RPE1 mRuby-PCNA cells expressing GFP-53BP1, which were manually analysed using Volocity software (PerkinElmer) as we were unable to achieve reliable tracking of 53BP1 foci, and hTert-RPE1 p21-GFP cells expressing CDK2L-mRuby (CDK2 sensor[28]) since there was no constant nuclear signal to track reliably.

For automated tracking of hTert-RPE1 cells expressing fluorescent markers in higher throughput we developed a cell-tracking pipeline in the commercial scripting language Matlab. The pipeline tracks cells and extracts data from sequences of uncompressed TIF images, captured from live cell imaging. Segmentation is based on well-established methods, while tracking is an implementation of a probabilistic tracking algorithm defined in refs 51,52. The code for cell tracking can be found online and contains an example (cellcycle.org.uk; Resources). The key steps in this pipeline are outlined below:

1. Nuclear segmentation:

   1. Initial background correction is performed on individual images; here a highly blurred image (very wide Gaussian kernel) is negated from the original image.
   2. Images from channels with a nuclear probe are combined to maximize the quality of nuclear segmentation, this principally involves channels containing the p21-GFP and mRuby-PCNA signal.
   3. Nuclei are subsequently segmented from the filtered, combined, image using *Marker Controlled Watershed Segmentation*.
   4. Segments with intensity or size below a threshold are filtered out, as these correspond to mis-segmented background noise.
   5. Segmented images are saved as equally sized zero matrices with each nuclei region uniquely labelled.

2. Feature extraction: features describing morphology and fluorescent marker levels and localization for each channel are obtained for each segmented object in each frame. Importantly, here we extract intensity and texture features from unfiltered images. The reasoning behind this is being that across the experiments we perform, average intensity levels can vary over several orders of magnitude for example, under Nutlin-3 treatment; plate-wise or well-wise filtering here can mask true levels of p21 relative to controls. Specifically the features we extract are:

   1. Cell area: area in pixels of the segmented nuclei
   2. Major axis length: length of the ellipse that is equivalent based on second order moments to the segment[53]
   3. Minor axis length: width of the ellipse that is equivalent based on second order moments to the segment[53]
   4. Eccentricity: eccentricity of the ellipse that is equivalent based on second order moments to the segment[53]
   5. Equivalent diameter: diameter of the circle with the same area as the segment
   6. Solidity: percentage of pixels in the convex hull surrounding the segment which are also in the segment
   7. Perimeter: perimeter length of the segment
   8. Roundness: defined as $4\pi \times$ area/perimeter$^2$ (ref. 53)
   9. PCNA mean intensity
   10. s.d. of PCNA intensity
   11. Kurtosis of PCNA intensity
   12. Foci strength: PCNA intensities below the mean are removed, the s.d. of pixels above the mean is then calculated. This identifies bright spots, and ignores dark spots in the nucleus, for example, nucleoli.
   13. p21 mean intensity
   14. p21 s.d.
   15. Mean silhouette score: $K$-mean clustering ($K = 2$) is performed on nuclei segments, and the mean silhouette score of the two clusters calculated. This feature identifies dumbbell shapes and is indicative of double nuclei segments. These features are defined in the single-cell track matrices presented online. In these matrices the first column gives cell index, the second time point and third migration distance. Subsequent columns are features as defined above. For CDK2 activity data, p21 intensities are

replaced with CDK2L-GFP nuclear intensity, CDK2L-GFP ring region intensity, and the ratio of these two values. The ring region, here defined as a seven pixel wide ring surrounding the nucleus, has been shown to perform effectively in measuring nuclear translocation[54].

3.  Segment probability: segments can be one of five defined objects, either correctly segmented (one nucleus, mitotic nucleus, nucleus exiting mitosis) or incorrectly segmented (no nuclei, more than one nuclei). Probabilistic tracking maximizes the likelihood that a track corresponds to the movements and divisions of a single nucleus. Thus for tracking, probabilities are assigned to each of the five possibilities mentioned, for each segment, in each frame such that tracking can maximize the likelihood a single nucleus is followed. This is performed by manually selecting examples of each of the defined objects, for example, mitotic nuclei, or multiple nuclei, from all segmented objects within a graphical user interface (GUI). Subsequently, a support vector machine classifier is trained on these manually selected examples. The classifier is used to assign a probability to every segment, in every frame, of it being one of these objects. As such, every segmented object, in every frame, is assigned five posterior probabilities corresponding to the likelihood of it being: one nucleus; mitotic nucleus; nucleus exiting mitosis; no nuclei; more than one nuclei.

4.  Probabilistic tracking: The goal of automated cell tracking is to generate a set of tracks that correspond to individual nuclei as they move across the field of view (FOV), and to capture cell division events, such that the lineage of nuclei is recorded. Briefly, in the automated probabilistic tracking algorithm used[51], hill-climbing optimization is performed to maximize a cost function designed to meet the goal set out above. Specifically the cost function penalizes large movements, and rewards tracks which follow segments that have a high probability of being a single cell; or an object with a high probability of being mitotic where in subsequent frames two objects exist that have a high probability of being mitotic exit cells. Optimization is performed by iteratively adding the highest scoring track, from the set of all possible new tracks, given the current set of tracks. To find the highest scoring track, given the current set of tracks, a dynamic programming technique is used. Notably, the algorithm can handle gaps and nuclei entering and exiting the FOV[51].

5.  Manual correction: automated tracking performs the majority of work in tracking the hTERT-RPE1 cells, however these cells are highly motile, and some of the conditions we use increase motility even further. As a result, the error rate in tracking ranges from 1 mistake, per cell, per 300 frames; to 1 mistake, per cell, per 20 frames. In the cell cycle analysis we are performing it is essential we obtain near 100% accuracy. As such, we developed a GUI to efficiently amend errors in automated tracking, as well as compare the feature values of tracked cells against the original confocal microscopy images. This allows us to achieve nearly 100% accuracy in tracking, and more importantly if we identify outliers, or unexpected results in later analysis we can return to the videos and visually inspect whether the result is genuine or the product of a tracking error.

**Defining G1/S and S/G2 transitions.** We define G1/S transition as the point at which both mRuby-PCNA intensity begins to increase and we see the appearance of textured PCNA in the nucleus. This is characterized by a small jump in the PCNA foci strength feature (Fig. 1b; Supplementary Fig. 2g). As further evidence this is in fact the G1/S transition, we utilized hTERT-RPE1 cells expressing both mRuby-PCNA and endogenously tagged CyclinA2-mVenus. Single-cell traces of this dual expressing line showed us at a single-cell level CyclinA2-mVenus intensity increases at exactly the same time as mRuby-PCNA intensity, providing further evidence this is the G1/S transition (Supplementary Fig. 2g).

We define the S/G2 transition as the point at which mRuby-PCNA foci are at their maximum strength (Fig. 1a); after this we see rapid disassembly of foci and homogenous PCNA levels across the nucleus. This definition is consistent with previous studies looking at nuclear PCNA dynamics[55]. Quantitatively, this time point is characterized by the peak of a sharp spike in the PCNA foci strength measure; this sharp spike being exhibited by all cells undergoing S-phase. At this time point we also see p21-GFP levels switch on in all cells that express p21-GFP in G2.

Mitosis was defined by gross morphology changes, principally this is seen as a spike in cell area lasting for one or two frames, corresponding to nuclear envelope breakdown, followed by the nuclear area becoming significantly smaller (Fig. 1b). All transitions were manually marked on cell tracks, for further analysis and computational synchronization of tracks, using a GUI tool developed in Matlab. Single-cell tracks and manually labelled time points for each figure are provided online as Matlab matrices, with scripts to visualize time point positions.

**Sampling cell tracks for figures and normalization.** To determine correlations, between p21 levels and phase length, for both control data and depletion experiments, a single-cell lineage was used. This was selected based on the daughter cell captured within the FOV for the longest period of time. The likelihood of arrest observed using this sampling regime very closely matched (21% $n = 47/219$) that observed for sampling both daughters, where only mitosis in which both daughters

are subsequently tracked for greater than 60 frames are used (23% $n = 29/128$). The single lineage-sampling regime is used outside of calculations comparing daughters, as here n is greater. In all conditions at least two technical replicates were analysed, and tracks from each replicate were visually compared to ensure dynamics were consistent, prior to pooling of the data.

Relative p21 levels decreased over imaging, this is due to intensity adjustments made by the microscope, due to increasingly bright images, (more p21-GFP expressing cells). No adjustment for this decrease was made, such that data remains faithful to the original imaging data sets, though baseline levels for arrest thresholds are defined at 15 a.u.

For all p21-GFP traces except Fig. 1b, p21-GFP levels were normalized using the lowest level of p21-GFP captured across all tracks. This value (4 a.u. for Fig. 1c traces) was generally similar to those seen in imaging (p21-GFP background levels were 12 a.u. In the first frame at the 10th percentile, while p21-GFP background levels were 2 a.u. in the last frame at the 10th percentile). The lowest track level was used rather than imaging baseline values, to avoid negative values.

**Maximum and mean p21 expression levels and P-values.** p21-GFP values, used for population mean, s.d. and correlation measures in the main text are derived from the maximum p21-GFP value reached within a single cell in the cell cycle phase defined. The maximum p21-GFP value was used rather than the mean p21 value, as this generally gave stronger correlations to cell cycle phase length, though in all cases analysed for both mean and maximum the mean value also provided a statistically significant correlation.

While both maximum G1 p21 levels and G1 length were not normally distributed as determined by a one sample Kolmogorov–Smirnov test (G1 p21 levels, $P = 1.8677\mathrm{e}\text{-}04$; G1 length, $P = 0.0012$). Following log transformation, the sample fell under the normal distribution (G1 p21 levels, $P = 0.1467$; G1 length, $P = 0.2408$). Therefore, we calculated the Pearson correlations on the log-transformed data as well, since Pearson's correlation does not fully describe the association of data which does not fall under a bivariate normal distribution. While correlation values did change, at no point did a previously significant result become insignificant. As such, for simplicity and to match that data presented in figures, correlations determined on the untransformed data are given in the main text.

Supplementary Table 1 gives the values quoted in the main text (derived from the maximum p21 value per cell, per phase) alongside the respected value derived from the mean p21 level per cell. Full P-values are also quoted in this table; otherwise $P < 0.01$ is marked by **, and $P < 0.05$ is denoted by *. Correlations between p21-GFP levels (mean and maximum) with phase lengths following S-phase are also quoted. Results following log transformation of the max p21 levels versus length data are also given.

**Mitotic timing masks correlation between G2^M and G1^D lengths.** Given that we observe strong correlation between i) p21-GFP levels in the mother G2 ($G2^M$) and daughter G1 ($G1^D$), ii) p21-GFP levels in $G2^M$ and $G2^M$ length, and iii) p21-GFP levels in $G1^D$ and $G1^D$ length; we were initially surprised to find no correlation between $G2^M$ length and $G1^D$ length. However, in this comparison noise in the timing of mitosis has the dual effect of increasing $G2^M$ length and decreasing $G1^D$ length, and vice versa. Thus, if noise in mitotic timing is indeed masking the relationship between cell cycle phase length and p21 levels, we reasoned that the combined length of $G2^M$ and $G1^D$ would correlate better with the maximum p21-level reached than either length alone. We found this to be the case with $R = 0.68^{**}$ for maximum p21 level in $G2^M/G1^D$ against length of $G2^M/G1^D$, versus against length of $G1^D$ ($R = 0.62^{**}$) or $G2^M$ ($R = 0.52^{**}$) alone. This result was calculated from the merged experimental repeats ($n = 4$). Within each repeat this result held as well (see Supplementary Table 2).

**Nutlin-3 leads to rate dependent G2 arrest.** In cells exiting S-phase following Nutlin-3 treatment we observed a G2 arrest that was dependent on the rate of p21 accumulation, meaning, those cells with the fastest rate of p21-GFP accumulation ($\mu = 0.25$ a.u. per min) at the end of S-phase arrested in G2, while those displaying a slower rate of p21-GFP accumulation ($\mu = 0.1$ a.u. per min) underwent mitosis and arrested in the following G1 (Fig. 2d). Furthermore, the rate of p21 increase between 100 and 200 min following S/G2 correlated with arrest ($R = 0.46^{**}$). Thus G2 arrest appears to be dependent on the rate of p21 accumulation following S/G2 transition. Finally, we note that those cells that were earlier in S-phase at the time of Nutlin-3 addition, are more likely to arrest compared to those exiting S-phase sooner, this is evident from correlation observed between S/G2 transition time and whether a cell undergoes G2 arrest or not ($R = 0.34^*$). We suggest that this is due to accumulated p53 stabilization over S-phase. This means that the longer a cell is in S-phase in the presence of Nutlin-3, more p53 is stabilized and more p21 mRNA accumulates. Thus when a cell exits S-phase and p21 degradation is switched off, p21 protein accumulates faster, increasing the likelihood of G2 arrest.

**S-phase entry is compromised following Cdt2 depletion.** Following Cdt2 depletion, or Cdt2 and Skp2 co-depletion, we observed significantly fewer cells entering S-phase. For each FOV we counted the number of traces obtained in

which cells are captured for an entire S-phase, and periods of 100 min before and 200 min after (as plotted in Fig. 5a). We found the average number of traces per FOV to be 32.5 for control siRNA, 27 for Skp2 siRNA, 12.3 for Cdt2 siRNA, and 6.3 for Cdt2 and Skp2 co-depletion.

**Mathematical modelling.** Based on the wiring diagram in Fig. 6a, we derived a set of nonlinear ordinary differential equations to describe the rate of change of components in the p21 control network over time (see Supplementary Software 2). Reaction kinetics were chosen according to the law of mass action with a few exceptions listed below.

Protein transcription and translation: Part of the noise in cell cycle progression stems from the low copy numbers of mRNAs that encode for regulatory proteins[56]. To account for this, we modelled the level of mRNAs of p21 ($mRNA_{p21}$), cyclins ($mRNA_{Cy}$), and p53 ($mRNA_{p53}$).

$$\frac{dmRNA_{p21}}{dt} = k_{mRNA}^{Sy} + k_{mRNA,p53}^{Sy} \cdot p53 - k_{mRNA}^{De} \cdot mRNA_{p21}, \quad (1)$$

$$\frac{dmRNA_{Cy}}{dt} = k_{mRNA}^{Sy} - k_{mRNA}^{De} \cdot mRNA_{Cy}, \quad (2)$$

$$\frac{dmRNA_{p53}}{dt} = k_{mRNA}^{Sy} - k_{mRNA}^{De} \cdot mRNA_{p53}, \quad (3)$$

where mRNAs are synthesized and degraded with the constitutive rates $k_{mRNA}^{Sy}$ and $k_{mRNA}^{De}$, respectively, and $k_{mRNA,p53}^{Sy}$ denotes the rate of p53-dependent synthesis of p21 mRNAs. Here and in the rest of the model we assumed that cyclin corresponds to the sum of CyclinE and CyclinA, which both activate CDK2. p21 ($p21$), cyclins ($Cy$) and p53 ($p53$) are produced from their mRNAs according to the following equations.

$$\frac{dp21_t}{dt} = k_{p21}^{Sy} \cdot mRNA_{p21} - r_{p21}^{De} \cdot p21_t, \quad (4)$$

$$\frac{dCy_t}{dt} = k_{Cy}^{Sy} \cdot mRNA_{Cy} - r_{Cy}^{De} \cdot Cy_t, \quad (5)$$

$$\frac{dp53}{dt} = k_{p53}^{Sy} \cdot mRNA_{p53} - r_{p53}^{De} \cdot p53, \quad (6)$$

with $k_{p21}^{Sy}$, $k_{Cy}^{Sy}$ and $k_{p53}^{Sy}$ denoting the protein synthesis rates, and $r_{p21}^{De}$, $r_{Cy}^{De}$ and $r_{p53}^{De}$ the corresponding degradation rates. The subscript t indicates total protein levels, which include the protein in its free from and as part of larger complexes. Also note that since CDK2 is in excess over its cyclins[57], $Cy_t$ corresponds directly to the total level of CDK2:Cyclin complexes with $Cy$ denoting the level of active CDK2:Cyclin. Protein degradation rates are as follows.

$$r_{Cy}^{De} = k_{Cy}^{De} + k_{Cy,Cy}^{De} \cdot Skp2 \cdot Cy, \quad (7)$$

$$r_{p21}^{De} = k_{p21}^{De} + k_{p21,Cy}^{De} \cdot Skp2 \cdot Cy + k_{p21,RCa}^{De} \cdot Cdt2 \cdot RC_a, \quad (8)$$

$$r_{p53}^{De} = \frac{k_{p53}^{De}}{j_{p53} + Dam}. \quad (9)$$

The constitutive rates of cyclin and p21 degradation are $k_{Cy}^{De}$ and $k_{p21}^{De}$, respectively. In addition, CDK2-mediated phosphorylation can promote cyclin degradation with rate $k_{Cy,Cy}^{De}$ in an Skp2-dependent manner[58–60]. Here, $Skp2$ denotes the relative level of the SCF$^{Skp2}$ ubiquitin ligase. Similarly, p21 degradation occurs with rate $k_{p21,Cy}^{De}$ through a pathway involving CDK2 phosphorylation and Skp2 (refs 24,59), but also with rate $k_{p21,RCa}^{De}$ via CRL4$^{Cdt2}$ (Cdt2), which is recruited to chromatin-bound PCNA[24,40] within aRCs ($RC_a$). For p53, stabilization has been shown to occur in response to DNA damage[61], which we accounted for by assuming that p53 degradation with rate $k_{p53}^{De}$ is inversely proportional to the sum of DNA damage (Dam) and an inhibition constant $j_{p53}$.

CDK2 inhibition. At high levels, p21 can act as an efficient inhibitor of CDK2 (ref. 6). Hence, we assumed that p21 binds to CDK2:Cyclin complexes converting them from an active form ($Cy$) into an inactive one ($CyP21$).

$$\frac{dCyP21}{dt} = k_{CyP21}^{As} \cdot p21 \cdot Cy - \left(k_{CyP21}^{Ds} + r_{p21}^{De} + r_{Cy}^{De}\right) CyP21, \quad (10)$$

$$\text{with } Cy = Cy_t - CyP21, \quad (11)$$

where $k_{CeP21}^{As}$ and $k_{CeP21}^{Ds}$ denote the association and dissociation rate constants, respectively, and the conservation equation (11) holds.

S-phase entry and DNA synthesis: Licensing of replication origins occurs during late mitosis or early G1-phase resulting in the formation of pre-replication complexes[62]. In our model, CDK2 activates these pre-replication complexes (RC) converting them into replication complexes that are primed for PCNA loading ($RC_p$).

$$\frac{dRC}{dt} = -r_{Rc}^{Ph} \cdot RC + k_{Rc}^{Dp} \cdot RC_p - r_{Rc}^{Ds} \cdot RC \quad (12)$$

$$\text{with } r_{Rc}^{Ph} = k_{Rc}^{Ph} \frac{Cy^n}{j_{Cy}^n + Cy^n}. \quad (13)$$

Here, priming occurs by CDK2 phosphorylation with rate $k_{Rc}^{Ph}$ in an ultrasensitive fashion with Hill coefficient $n$ at a CDK2:Cyclin threshold of $j_{Cy}$. Primed replication complexes can revert back to an unprimed state with a small rate $k_{Rc}^{Dp}$ and replication complexes are disassembled upon completion of S-phase with rate $r_{Rc}^{Ds}$ (see below). Once primed, replication complexes can bind free PCNA ($PCNA_a$) or PCNA with a p21 molecule bound to its PIP-box ($PCNA_i$) forming either active ($RC_a$) or inactive ($RC_i$) replication complexes, respectively.

$$\frac{dRC_p}{dt} = r_{Rc}^{Ph} \cdot RC - k_{Rc}^{Dp} \cdot RC_p - k_{RcPc}^{As}$$
$$\cdot (PCNA_a + PCNA_i)RC_p + k_{RcPc}^{Ds}(RC_a + RC_i) - r_{Rc}^{Ds} \cdot RC_p, \quad (14)$$

$$\frac{dRC_a}{dt} = -k_{PcP21}^{As} \cdot p21 \cdot RC_a + \left(k_{PcP21}^{Ds} + r_{p21}^{De}\right)RC_i + k_{RcPc}^{As} \cdot PCNA_a$$
$$\cdot RC_p - k_{RcPc}^{Ds} \cdot RC_a - r_{Rc}^{Ds} \cdot RC_a, \quad (15)$$

$$\frac{dRC_i}{dt} = k_{PcP21}^{As} \cdot p21 \cdot RC_a - \left(k_{PcP21}^{Ds} + r_{p21}^{De}\right)RC_i + k_{RcPc}^{As} \cdot PCNA_i$$
$$\cdot RC_p - k_{RcPc}^{Ds} \cdot RC_i - r_{Rc}^{Ds} \cdot RC_i, \quad (16)$$

where $k_{RcPc}^{As}$ and $k_{RcPc}^{Ds}$ correspond to the rates of PCNA loading and unloading, respectively. Note that p21 is the strongest known binding partner of PCNA[63] and it was proposed that it competes with other PCNA-binding proteins such as processivity factors of DNA synthesis[42,64]. Thus, we assumed that binding of p21 to aRCs with rate $k_{PcP21}^{As}$ (or binding of PCNA:p21 complexes to $RC_p$) creates inaRCs[65,66]. Dissociation of p21 from PCNA in these complexes occurs with rate $k_{PcP21}^{Ds}$. In our model, aRCs synthesize DNA (DNA) and once DNA replication is finished, replication complexes disassemble with rate $r_{Rc}^{Ds}$.

$$\frac{dDNA}{dt} = k_{Dna}^{Sy} \cdot RC_a, \quad (17)$$

$$r_{Rc}^{Ds} = H(DNA - 1). \quad (18)$$

Here, $k_{Dna}^{Sy}$ is the DNA synthesis rate and $H$ the Heaviside function.

PCNA dynamics: We assumed that free PCNA in the nucleus ($PCNA_a$) is replenished from a cytoplasmic pool and that p21 can bind to PCNA creating PCNA:p21 complexes ($PCNA_i$).

$$\frac{dPCNA_a}{dt} = k_{Pc}^{Im} - k_{PcP21}^{As} \cdot p21 \cdot PCNA_a + \left(k_{PcP21}^{Ds} + r_{p21}^{De}\right)PCNA_i$$
$$- \left(k_{Pc}^{Ex} + k_{RcPc}^{As}RC_p\right)PCNA_a + \left(k_{RcPc}^{Ds} + r_{Rc}^{Ds}\right)RC_a, \quad (19)$$

$$\frac{dPCNA_i}{dt} = k_{PcP21}^{As} \cdot p21 \cdot PCNA_a - \left(k_{PcP21}^{Ds} + r_{p21}^{De}\right)PCNA_i$$
$$- \left(k_{Pc}^{Ex} + k_{RcPc}^{As}RC_p\right)PCNA_i + \left(k_{RcPc}^{Ds} + r_{Rc}^{Ds}\right)RC_i, \quad (20)$$

where $k_{Pc}^{Im}$ and $k_{Pc}^{Ex}$ denote the nuclear import and export rates of PCNA, respectively. The association and dissociation of p21 and free PCNA occur with rate $k_{PcP21}^{As}$ and $k_{PcP21}^{Ds}$, respectively, while the binding and dissociation rates of PCNA and replication complexes are $k_{RcPc}^{As}$ and $k_{RcPc}^{Ds}$, respectively. Considering all the complexes that contain p21 the following conservation equation holds.

$$p21 = p21_t - CyP21 - PCNA_i - RC_i. \quad (21)$$

DNA damage. In our model, DNA damage (Dam) can either occur randomly, independent of the cell cycle stage, or specifically during DNA replication.

$$\frac{dDam}{dt} = k_{Dam}^{Ge} + k_{Dam,RCa}^{Ge}RC_a - r_{Dam}^{Re} \cdot Dam, \quad (22)$$

$$\text{with } r_{Dam}^{Re} = k_{Dam}^{Re} + k_{Dam,p53}^{Re} \frac{p53}{j_{Dam} + Dam}. \quad (23)$$

Here, $k_{Dam}^{Ge}$ and $k_{Dam,RCa}^{Ge}$ correspond to the constitutive and replication-dependent rate of DNA damage induction, respectively. Damage repair occurs with a p53-independent rate $\left(k_{Dam}^{Re}\right)$ and a p53-dependent rate $\left(k_{Dam,p53}^{Re}\right)$ assuming that p53 expression induces DNA repair processes.

Computation. A deterministic version of the model was prepared using the Systems Biology Toolbox 2 (ref. 67) for MatLab (version 8.6.0 R2015b) and simulated with the CVODE routine from SUNDIALS[68]. Bifurcation diagrams were calculated using the freely available software XPP-Aut[69]. Different versions of the model are available at http://cellcycle.org.uk/publication. The model was also deposited in the BioModels database[70] and assigned the identifier MODEL1607210001. Kinetic parameters were estimated by comparing deterministic simulations to experimental time-course data of cell cycle regulators and to qualitative experimental observations. Parameter values and non-zero initial

conditions are listed in Supplementary Tables 3 and 4. We simulated a stochastic version of the model using custom-made MatLab code of the stochastic simulation algorithm, also known as Gillespie's algorithm (reviewed in ref. 71) according to the sorting direct method[72]. To this end, the rate expressions of the deterministic model were converted into propensity functions, that requires the transformation of the relative levels of cell cycle regulators into numbers of molecules. Here, we followed the system in a control volume containing on average 1,000 molecules per AU of proteins, 100 molecules per AU of mRNAs and 1 event per AU of DNA damage. Hence, most of the simulated cell-to-cell variability originates in the stochastic infliction of DNA damage followed by transcriptional noise, while translational noise plays only a minor role. To account for the extrinsic noise observed in our experiments, that is, the variability in p21 levels after cell division, we multiplied the initial p21 level, the initial p21 mRNA level and the p53-independent p21 mRNA synthesis rate ($k_{mRNA}^{Sy}$ in equation (1)) for each cell (simulation run) by a single random number drawn from a uniform distribution in the interval (0.1,1). For the simulation of depletion experiments Skp2 and/or Cdt2 were reduced to 1%.

**Code.** Newly generated code used for cell tracking and feature extraction, and for generating our mathematical model is freely available at http://cellcycle.org.uk/publication

**Data availability.** The data sets generated during the current study are available from the corresponding authors on reasonable request.

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

## Acknowledgements

We thank Feng Zhang for pX330-U6-Chimeric_BB-CBh-hSpCas9 (Addgene plasmid #42230) and Jiri Lukas for GFP-53BP1. We thank Jon Pines for critical reading of the manuscript and the ICR FACS facility for technical support. The groups of C.B. and B.N. are funded by a BBSRC Strategic LoLa grant (BB/M00354X/1). J.M. is funded by a DFG Emmy Noether grant (MA 5831/1-1).

## Author contributions

Conceptualization, A.R.B.; Investigation, A.R.B., F.B., H.S.; Resources, J.M.; Methodology, Validation, Formal Analysis, A.R.B., S.C., F.S.H.; Software, S.C., F.S.H.; Writing–Original Draft & Reviewing and Editing, A.R.B., S.C., F.S.H., C.B., B.N.; Funding Acquisition, C.B., B.N.

## Additional information

**Competing interests:** The authors declare no competing interests.

