## [Peer Review File · Nature Communications]

Reviewers' Comments:

Reviewer #1 (Remarks to the Author)

In the presented paper by Barr et al. the authors apply a single cell quantitative approach to investigate how mammalian cells integrate the DNA damage signals in order to make a decision to divide or arrest. The authors propose that cellular levels of the cell cycle inhibitor p21 reflect the history of DNA damage in the cell's G1 phase and its mother's G2 phase. These levels define whether the cell is going to proceed through the cell cycle or arrest. If the cell decides to divide, the p21 levels are reset to zero through the SCF- and CRL4-dependent degradation of p21 in late G1 and S.

In summary, the authors verify at single cell level some principles that were previously studied in bulk experiments. They demonstrate that DNA damage induces p53-dependent expression of p21; that high levels of p21 can cause cell cycle arrest; and that p21 is degraded in late G1 phase, stays low during the S phase and then starts to increase in G2, M and G1 of the following cycle. The main conceptual novelty of this paper is the idea that p21 levels are used by the cells to integrate and store the information about DNA damage signaling history during G2 phase and the following G1. Thus, by modulating p21 levels the DNA damage may cause cells to arrest in G1. Overall, this is an interesting concept but unfortunately I don't sufficient experimental validation of this idea in the paper. On the contrary, some of the presented data seem to contradict to this idea. For example, G1 length is uncorrelated with p21 at endogenous levels in cycling cells. Therefore, this manuscript is not appropriate for publication in this journal.

Major points:

- 1) The authors propose that cells use p21 levels to integrate p21 signals over G2 and G1 phases, however they only studied how cells respond to DNA damage that occurs in S phase (endogenous levels of replicative stress, or additional S-phase damage induced by drugs). Applying other chemicals or irradiation would help to look at DNA damage responses occurring outside S phase.
- 2) In the Discussion section (e.g., Fig 7F and Discussion on page 22) the authors propose that intermediate levels of DNA damage should cause intermediate p21 increase and ultimately the G1 delay for DNA repair. However, when analyzing the results, the authors conclude that the G1 length is not influenced by p21 levels (Fig 2A,B). Rather, they conclude that p21 levels are interpreted by the cell in a binary way (to divide or arrest), and cannot cause G1 elongation for DNA repair. This result looks surprising and counterintuitive in the framework where you consider p21 levels as the main readout for a cumulative amount of DNA damage.
- 3) The central proposal that p21 levels are determined by the integration of DNA damage signaling events during G2(mother) and G1(daughter) is not sufficiently supported by data and is based mainly on the correlation analysis of p21 levels and G2(mother)+G1(daughter) timing (page 7). More direct evidence is required to prove this hypothesis.
- 4) A large increase in p21 levels during G1 compared to the G2 phase of the previous cell cycle suggests that p21 levels are strongly modulated by the degradation machinery, not just by the synthesis induction rate. The authors are only considering the synthesis aspect, however, the observed heterogeneity in p21 levels between the cells and within one cell over time could well be mediated by differences in degradation rates, which are not measured. Furthermore, for p21 to keep memory about the events occurring during G2 and G1, it should have a sufficiently long half life period during these phases. To support their idea, the authors should measure p21 stability.

Reviewer #2 (Remarks to the Author)

Review of Barr et al., Nat Comms.

This is an original and interesting paper on the role of p21 in cell fate decisions in non-transformed cells, which demonstrates nicely several useful findings. Among these are that levels of p21 in single cells correlate with endogenous DNA damage, that p21 is absent in S-phase due to bistable Cdt2-dependent degradation, that eliminating p21 largely prevents cellular quiescence but does not alter the length of G1 (which is shown to be p53-dependent), that preventing p21 degradation is associated with DNA damage and cell death, and that altering p21 levels slightly alters PCNA levels and nuclear / cytoplasmic localisation. Other secondary findings include that p21 is not degraded in mitosis, correcting somewhat marginal data previously published by Michele Pagano's group in 2007, and that endogenous DNA damage, where it occurs, is insufficient to arrest cells in G2 but rather arrests them in G1. Among the strong points of the paper are: that the analysis is achieved by single cell analysis in asynchronous culture without using synchronising agents that perturb cell cycle control; a quantitative approach based on high content analysis of information from a CRISPR-mediated knockin of p21-GFP on both alleles (ie, strong methodology, that in itself would make a useful contribution to the literature); a mathematical model for control of p21 degradation that fits the data and allows insights into G1 control (ie a nice integration of systems biology and cell biology).

Among the not-so-strong points are the insufficient characterisation of the p21-GFP behaviour (point 1, below), a somewhat confusing writing style (point 2), a slight technical issue with the CDK2 activity reporter (point 3), some unexplained confusing data (point 4), a lack of rescue experiments or workarounds in knockout or knockdown experiments (point 5), some assertions that are not well justified (point 6), and an unhelpful title (point 7).

On the whole, I think, with some improvements, that this paper could make a nice addition to the literature and would merit publication in a high profile journal. I would therefore recommend that the manuscript should be extensively revised before publication.

1. The p21-GFP construct and cells should be better described: Does it still bind to and inhibit CDKs? This cannot be ascertained from the inverse correlation of CDK2 activity with p21 levels since (a) this is just a correlation: CDK activity is controlled largely by cyclin degradation; (b) it is shown that CDK2 activity can rise before p21 levels have dropped. Is variability of expression of p21-GFP reflected by variability of endogenous p21 levels in WT cells? Can changes in p21 throughout the cell cycle that are detected by GFP-fluorescence also be seen by western blotting? What happens to p21-GFP levels upon serum starvation or cell cycle reentry?

2. The writing is not always clear, and this makes it difficult to appreciate the importance of the results given the vast literature on the subject. For example, playing devil's advocate, one could say that it is already well known that p21 levels decrease in S-phase, that they are controlled by Skp2 and Cdt2, that replication stress causes p21 levels to increase, that p21 knockdown hinders cell cycle exit, and so on. The novelty of the results, as opposed to the methodology or the nice hypotheses (memory, resetting, etc), is not sufficiently highlighted. One of the most interesting points for me, that altering p21 levels in G1 do not affect G1 length, whereas p53 seems to do so, is not evident from the abstract.

3. The CDK2 reporter described still works normally in CDK2 knockout cells (our unpublished data), so it is not necessarily specific to CDK2. CDK1 substitutes for CDK2 in the knockouts. While it is likely that CDK2 activity is the main activity in WT cells in G1, this caveat should be addressed or it should be mentioned that this reflects combined CDK1/2 activity.

4. I don't understand the differences in height of the p21 peaks on the left and right in two

successive G1 phases in single cells (Fig 1B, and above all, fig 1C, compared with, for example, fig 5A, where peaks are of similar size). Is this a technical issue? Does live imaging of GFP-fluorescent cells itself generate DNA damage, leading to higher levels throughout the experiment? In Fig 2G it is claimed that CDK2 activity is low, but the units are arbitrary, so how can this be known? Although the scale is, as stated, different from 2F, if one takes the absolute values, it looks like CDK2 activity in the cell shown in 2G is around 20au, which corresponds to the high level of 2F. I also don't understand why PCNA levels are higher in p21 KO cells (Fig 7B) but lower upon DNA damage in P21 knockdown cells (Fig 7D) - why was this experiment not done in KO cells?

5. As I understand it, siRNA experiments, or knockouts, if not rescued by si-resistant constructs or re-expression, should be supported by independent lines of evidence that the effects are specific to the genes targeted. For example, nutlin addition could counteract the effect of p53 siRNA on G1 length (fig 2A) – incidentally, the text refers to fig 2C as showing that nutlin causes G1 arrest, but it is not clear to me how this is inferred from the data shown. Additionally, p21 siRNA could be performed in p21 KO cells, which should show that the effects on PCNA levels are eliminated.

6. It is stated in the text on p15 and in the legend to figure 5 that p21 degradation is essential for S-phase entry. While the correlation is certainly correct, this is not strictly proven, since co-depletion of Cdt2 and Skp2 apparently did not prevent S-phase entry in all cells, even though those that entered S-phase had undetectable p21. This could be simply because p21 degradation is PCNA-dependent, so there is negative feedback. Co-depletion of p21, as performed with Cdt2 depletion alone, would have helped: this should rescue S-phase entry. But still this would not prove that the low levels of p21 in G1 prevent S-phase entry. Another slightly contentious assertion is that high levels of p21 are required to inhibit CDK2 activity (see also point 4). At no point is this formally proven.

7. The title, while being commendably catchy, does not really convey the experimental findings. It is somewhat debatable that this is really “memory”. Besides, while it is a nice idea that could be well discussed in the appropriate section, I think it would require further experimental investigation. If there is such a thing as DNA damage memory, to prove that it is “encoded” by p21 rather than something else (eg, the sum-total of phosphorylation of Chk1 substrates), would require much further investigation. I do not have any brilliant ideas about what the title of the paper could be. Nevertheless, I recommend that the authors consider how to better represent the key findings of the paper with clarity in the title rather than trying to make it sound too conceptual, at the risk of the title being shown to be incorrect after future studies, which would be a shame.

Minor points

1. There are many ambiguous sentences due to grammatically incorrect absence of the word “that”. E.g. in the abstract: “We show p21 degradation by the combined actions of the10 ubiquitin ligases SCFSkp2 and CRL4Cdt2 leads to irreversible S-phase entry”. This should be “We show THAT...”

2. P10, line 4, refers to (Figs 2A, S1B): “we saw no change in G1 length distribution (Figs 2A;S1B)”. However, S1B does not present any data on this.

Reviewer #1 (Remarks to the Author):

The main conceptual novelty of this paper is the idea that p21 levels are used by the cells to integrate and store the information about DNA damage signaling history during G2 phase and the following G1. Thus, by modulating p21 levels the DNA damage may cause cells to arrest in G1. Overall, this is an interesting concept but unfortunately I don't sufficient experimental validation of this idea in the paper. On the contrary, some of the presented data seem to contradict to this idea. For example, G1 length is uncorrelated with p21 at endogenous levels in cycling cells. Therefore, this manuscript is not appropriate for publication in this journal.

We thank this Reviewer for their comments. What is clear from their summary is that we have not been sufficiently clear in the manuscript as to what our aims and objectives were and why our results are novel and lead to an advance in the field. We have made extensive revisions to the manuscript to make it obvious why our work will have an impact in the field of cell cycle biology. We hope the reviewer will now appreciate how novel this study is.

Major points:

1) The authors propose that cells use p21 levels to integrate p21 signals over G2 and G1 phases, however they only studied how cells respond to DNA damage that occurs in S phase (endogenous levels of replicative stress, or additional S-phase damage induced by drugs). Applying other chemicals or irradiation would help to look at DNA damage responses occurring outside S phase.

We apologise if there is some confusion over our hypotheses and how we tested them. Our data collected from cells not exposed to chemicals or irradiation demonstrated that DNA damage is occurring in a sub-population of normal proliferating cells. We hypothesised that this DNA damage was occurring during S-phase and confirmed this using drugs that induce replicative stress (i.e. aphidicolin and ATR inhibitor). How p21 is regulated in different cell cycle phases following exposure of cells to DNA damaging agents, such as chemicals or irradiation, has long been studied by other groups and was not the focus of our work. In fact, we believe that our work here better represents how p21 is regulated in single cells in physiologically-relevant contexts, rather than in situations where cells are bombarded with DNA damaging agents. We have now revised the text in the introduction (starting p3) and the results to make this point clear (e.g. p13, lines 288-289).

2) In the Discussion section (e.g., Fig 7F and Discussion on page 22) the authors propose that intermediate levels of DNA damage should cause intermediate p21 increase and ultimately the G1 delay for DNA repair. However, when analyzing the results, the authors conclude that the G1 length is not influenced by p21 levels

(Fig 2A,B). Rather, we conclude that p21 levels are interpreted by the cell in a binary way (to divide or arrest), and cannot cause G1 elongation for DNA repair. This result looks surprising and counterintuitive in the framework where you consider p21 levels as the main readout for a cumulative amount of DNA damage.

We have now made substantial revisions throughout our manuscript in order to make our findings clear. We have also updated the previous Fig. 7F, now Fig. 7c, to clarify the role of p21.

3) The central proposal that p21 levels are determined by the integration of DNA damage signaling events during G2(mother) and G1(daughter) is not sufficiently supported by data and is based mainly on the correlation analysis of p21 levels and G2(mother)+G1(daughter) timing (page 7). More direct evidence is required to prove this hypothesis.

Additional evidence in support of our specific model is provided by our experiments using aphidicolin that induces DNA damage during S-phase in the mother cell. Here, we observe an increase in both G2 and G1 p21-GFP levels and an increase in the proportion of cells entering G1pm arrest (Fig. 4b).

4) A large increase in p21 levels during G1 compared to the G2 phase of the previous cell cycle suggests that p21 levels are strongly modulated by the degradation machinery, not just by the synthesis induction rate. The authors are only considering the synthesis aspect, however, the observed heterogeneity in p21 levels between the cells and within one cell over time could well be mediated by differences in degradation rates, which are not measured.

This is an excellent point raised by the reviewer. Indeed, our mathematical model incorporates predicted rates for both p21 synthesis (via p53-mediated transcription) and degradation (through both Cdt2- and Skp2-dependent pathways) in one cell over time. Because this model accurately predicts p21 dynamics observed in both normal cells and following Cdt2- and/or Skp2-depletion experiments, we can be confident that these rates are well approximated. In fact, that we can predict these rates is another advantage of mathematical modelling. Our model supports our experimental observations suggesting that DNA damage is a major contributor to both inter- and intra-cellular heterogeneity in p21 protein, and we have now added a sentence into our results to stress this point (p17, starting line 404; and p19, starting line 432).

Furthermore, for p21 to keep memory about the events occurring during G2 and G1, it should have a sufficiently long half life period during these phases. To support their idea, the authors should measure p21 stability.

After revising the manuscript in light of both Reviewers' comments, we have decided to remove our memory hypothesis in favour of highlighting the novelty of our results.

Reviewer #2 (Remarks to the Author):

Review of Barr et al., Nat Comms.

Among the strong points of the paper are: that the analysis is achieved by single cell analysis in asynchronous culture without using synchronising agents that perturb cell cycle control; a quantitative approach based on high content analysis of information from a CRISPR-mediated knockin of p21-GFP on both alleles (ie, strong methodology, that in itself would make a useful contribution to the literature); a mathematical model for control of p21 degradation that fits the data and allows insights into G1 control (ie a nice integration of systems biology and cell biology)..... On the whole, I think, with some improvements, that this paper could make a nice addition to the literature and would merit publication in a high profile journal. I would therefore recommend that the manuscript should be extensively revised before publication.

We thank the Reviewer for supporting publication of this work in *Nature Communications*.

Among the not-so-strong points are the insufficient characterisation of the p21-GFP behaviour (point 1, below), a somewhat confusing writing style (point 2), a slight technical issue with the CDK2 activity reporter (point 3), some unexplained confusing data (point 4), a lack of rescue experiments or workarounds in knockout or knockdown experiments (point 5), some assertions that are not well justified (point 6), and an unhelpful title (point 7).

We thank this Reviewer for their suggestions and we have addressed them in the extensively revised manuscript.

Specific points raised by the Reviewer:

1. The p21-GFP construct and cells should be better described: Does it still bind to and inhibit CDKs?

Yes. In our hTert-RPE1 p21-GFP/mRuby-PCNA cell lines, p21 is tagged at both alleles, i.e. all p21 protein present in the cell is tagged with GFP. In our p21siRNA and p21 knockout experiments, cells without p21 cannot enter G1pm arrest (Fig. 2b). Since cells with tagged p21-GFP can still enter G1pm arrest, this provides evidence that p21-GFP is functional and can inhibit (and thus must be binding) CDKs. We have now included a description of this in the text (p5, starting line 92).

Further evidence is that the p21-GFP/mRuby-PCNA cell lines behaves identically to the mRuby-PCNA cell line in: (i) growth (Supplementary Fig. 1d); (ii) cell cycle phase length (Supplementary Fig. 1e); and (iii) proportion of cells entering G1pm arrest (Fig. 2b:

compare Con siRNA (p21-GFP/mRuby-PCNA cells) vs Wild-type (mRuby-PCNA cells).

To be certain that p21-GFP still binds to CDK2, we had performed an immunoprecipitation of p21-GFP and probed for CDK2. CDK2 binds to p21-GFP and we have now included a western blot showing this result in Supplementary Fig. 1f.

Is variability of expression of p21-GFP reflected by variability of endogenous p21 levels in WT cells?

Yes. We immunostained mRuby-PCNA hTert-RPE1 cells with a p21 antibody to detect endogenous p21 protein and we see the same heterogeneity in G1 and G2 cells. We have now included this analysis (Supplementary Fig. S2a).

Can changes in p21 throughout the cell cycle that are detected by GFP-fluorescence also be seen by western blotting?

It is well-established that p21 is expressed in G1 and G2 and degraded in S-phase and we have now included references to these published works in the results to demonstrate that our data is consistent with expression of endogenous p21 protein (p6, line 107). Immunostaining of endogenous p21 in unperturbed mRuby-PCNA hTert-RPE1 cells also shows that p21 expression is limited to G1 and G2 cells and is absent in S-phase, consistent with our p21-GFP live cell data. We have now included a figure demonstrating this (Supplementary Fig. 2a).

We question whether western blotting should be considered the “gold standard” for p21 dynamics. Western blotting experiments provide information only about the average expression of a protein in a population and cannot be used to understand the inter- and intra-cellular heterogeneity that exists in the levels of many proteins - especially p21. Furthermore, western blotting would require synchronising cells by serum starvation and, in such experiments, cells enter (and thus exit) S-phase at different times.

What happens to p21-GFP levels upon serum starvation or cell cycle reentry?

This is a very interesting question raised by the Reviewer and we thank them for raising this point.

We made two observations that suggest that G1pm arrest in response to serum starvation is p21-independent. First, if we quantify single cell levels of p21-GFP after serum starvation, we see that p21-GFP levels increase in some, but not all, cells arresting in G1. Second, in response to serum starvation, p21KO cells can still arrest in G1 as efficiently as p21WT cells. These data raise a very important point, not previously appreciated - that not all G1pm arrests are equal at the molecular level. In this work, we have characterised a p21-dependent G1pm arrest in response to endogenous DNA damage when serum is present. This is in contrast to a p21-independent G1pm arrest

after serum withdrawal.

We have now replaced the original Fig. 7 with these data and included a new results section describing these findings (p19, starting line 436) that we think will be interesting to a broad readership.

2. The writing is not always clear, and this makes it difficult to appreciate the importance of the results given the vast literature on the subject. For example, playing devil's advocate, one could say that it is already well known that p21 levels decrease in S-phase, that they are controlled by Skp2 and Cdt2, that replication stress causes p21 levels to increase, that p21 knockdown hinders cell cycle exit, and so on. The novelty of the results, as opposed to the methodology or the nice hypotheses (memory, resetting, etc), is not sufficiently highlighted. One of the most interesting points for me, that altering p21 levels in G1 do not affect G1 length, whereas p53 seems to do so, is not evident from the abstract.

We have extensively revised the manuscript in line with the Reviewer's suggestions such that readers can appreciate the importance and novelty of our results.

3. The CDK2 reporter described still works normally in CDK2 knockout cells (our unpublished data), so it is not necessarily specific to CDK2. CDK1 substitutes for CDK2 in the knockouts. While it is likely that CDK2 activity is the main activity in WT cells in G1, this caveat should be addressed or it should be mentioned that this reflects combined CDK1/2 activity.

We have revised the text (p10, starting line 207) to address this point.

4. I don't understand the differences in height of the p21 peaks on the left and right in two successive G1 phases in single cells (Fig 1B, and above all, fig 1C, compared with, for example, fig 5A, where peaks are of similar size). Is this a technical issue?

The reason that peaks seem to increase in height during the imaging period in Figs 1b and c is due to a plotting bias. In Fig. 1c, individual cell traces are aligned to the G1/S transition, therefore cells that are in G1pm arrest at the start of filming and do not enter S-phase during the imaging period are not plotted. However, if tracked cells then enter G1pm arrest in the next imaged cycle, p21-GFP will increase in these cells - this is plotted, and thus peak height appears to increase overall.

The difference between Figs 1c and 5a is also due to different plotting of the data – Fig. 1c is aligned to the S/G2 transition and Fig. 5a is aligned to the G1/S transition. If we take Fig. 1c and now align the data to the G1/S transition (**Figure A**, below, right-hand graph, below), the data now look similar.

Does live imaging of GFP-fluorescent cells itself generate DNA damage, leading to higher levels throughout the experiment?

No. We tested that imaging did not induce DNA damage by using different 488nm laser exposures (Fig. 1d) and found that imaging does not induce DNA damage or differences in p21-GFP levels. To stress this, we have now included an additional figure (Fig. 5c) showing that even under long-term imaging conditions (up to 2400 mins/40 hr), p21-GFP levels do not increase overall in control siRNA-depleted cells.

In Fig 2G it is claimed that CDK2 activity is low, but the units are arbitrary, so how can this be known? Although the scale is, as stated, different from 2F, if one takes the absolute values, it looks like CDK2 activity in the cell shown in 2G is around 20au, which corresponds to the high level of 2F.

We apologise if we did not display our data in an intuitive fashion and have now revised the figure in light of these comments. We have clarified the labelling on these graphs by colour-coding the axes in Figs 2f and 2g. In Fig. 2g, the CDK2 activity is actually plotted on the left-hand axis and so is at approximately 0.3 (the 20 a.u. refers to the p21-GFP level). The CDK2 activity scale is the same in Figs 2f and g, only the p21-GFP axis differs between the two.

In terms of the units of CDK2 activity, activity is calculated as a ratio (Cytoplasmic CDK2/Nuclear CDK2 = CDK2 activity). Therefore, the units are not strictly arbitrary and are internally standardised for each cell and experiment.

I also don't understand why PCNA levels are higher in p21 KO cells (Fig 7B) but lower upon DNA damage in P21 knockdown cells (Fig 7D) - why was this experiment not done in KO cells?

After revising the manuscript in light of the Reviewers' other comments, we have decided to remove this section on PCNA localisation changes in favour of experiments

examining the role of p21 in response to serum withdrawal.

5. As I understand it, siRNA experiments, or knockouts, if not rescued by si-resistant constructs or re-expression, should be supported by independent lines of evidence that the effects are specific to the genes targeted. For example, nutlin addition could counteract the effect of p53 siRNA on G1 length (fig 2A) – incidentally, the text refers to fig 2C as showing that nutlin causes G1 arrest, but it is not clear to me how this is inferred from the data shown. Additionally, p21 siRNA could be performed in p21 KO cells, which should show that the effects on PCNA levels are eliminated.

It is highly unlikely that siRNA and two independent p21 knockouts (targeted with two different guide RNA constructs) would have the same off-target effects. In fact, the siRNA and two different p21KO lines (p21KO1 and p21KO2) are independent lines of evidence that the phenotypes observed are specific to p21 removal from the system. We have now clarified in the results that we have used three independent ways of reducing p21 protein (p9, starting line 190).

We do not see the logic behind the suggested Nutlin and p21 siRNA experiments. First, since Nutlin stabilises p53 protein, Nutlin treatment in p53-depleted cells should have no effect, i.e. cells should not accumulate p21 and should not arrest. We would not predict it to counteract the effect of p53 siRNA on G1 length. Second, in p21KO cells, PCNA levels are higher than in WT cells, indicating that in the absence of p21, PCNA levels increase. Since p21 is absent in these cells, we do not understand how adding p21 siRNA into p21KO cells would lead to a reduction in PCNA levels.

The y-axis on Fig. 2c was incorrectly labelled on our initial submission and should have read “0, 500, 1000 and 1500 mins” (as opposed to “0, 50, 100 and 150 mins”). This has now been corrected and thus hopefully makes it clear there is a G1 arrest after Nutlin-3 treatment (i.e. cells are in G1 for >600 mins) with high p21 levels.

6. It is stated in the text on p15 and in the legend to figure 5 that p21 degradation is essential for S-phase entry. While the correlation is certainly correct, this is not strictly proven, since co-depletion of Cdt2 and Skp2 apparently did not prevent S-phase entry in all cells, even though those that entered S-phase had undetectable p21. This could be simply because p21 degradation is PCNA-dependent, so there is negative feedback. Co-depletion of p21, as performed with Cdt2 depletion alone, would have helped: this should rescue S-phase entry. But still this would not prove that the low levels of p21 in G1 prevent S-phase entry.

We agree and have edited the text to reflect the fact that we have not strictly proven this point and also to focus on the novelty of our results showing that Skp2 and Cdt2 degrade p21 with different rates and timings (p15, starting line 339).

Another slightly contentious assertion is that high levels of p21 are required to inhibit CDK2 activity (see also point 4). At no point is this formally proven.

As mentioned above, there was some misunderstanding with regards to point 4 and the scales for CDK2 activity that has now been corrected and hopefully helps to address this point.

In addition, there are other lines of evidence, which we agree are still correlative, but that provide strong support to the statement that high p21 is required to inhibit CDK2 activity. First, in Fig. 2e we show that depletion of p21 leads to higher CDK2 activity after mitosis, consistent with Spencer *et al.*, Cell 2013. Second, in Fig. 4 we show that in cells arresting in G2 following aphidicolin treatment, where p21-GFP levels significantly increase (Fig. 4b), CDK2 activity is also inhibited after 600 minutes (Fig. 4d) and p21 depletion is required for G2 arrest in response to aphidicolin. Thus, CDK inhibition occurs following G2 and G1 arrest, where both arrests are dependent on p21. Furthermore, in no case in either DMSO or aphidicolin-treated cells do we observe cells arresting with low p21 levels or high CDK2 levels.

7. The title, while being commendably catchy, does not really convey the experimental findings. It is somewhat debatable that this is really “memory”. Besides, while it is a nice idea that could be well discussed in the appropriate section, I think it would require further experimental investigation.... Nevertheless, I recommend that the authors consider how to better represent the key findings of the paper with clarity in the title rather than trying to make it sound too conceptual, at the risk of the title being shown to be incorrect after future studies, which would be a shame.

We thank the reviewer for their comment and we have changed the title to “DNA damage during S-phase mediates the proliferation-quiescence decision in the subsequent G1 via p21 expression” to better reflect our results. In addition, we have removed all references to our memory hypothesis, with only one final paragraph in the discussion speculating about p21 acting to integrate p53 pulsatile signalling.

Minor points

1. There are many ambiguous sentences due to grammatically incorrect absence of the word “that”. E.g. in the abstract: “We show p21 degradation by the combined actions of the ubiquitin ligases SCFSkp2 and CRL4Cdt2 leads to irreversible S-phase entry”. This should be “We show THAT...”

We have corrected this throughout the manuscript.

2. P10, line 4, refers to (Figs 2A, S1B): “we saw no change in G1 length distribution (Figs 2A;S1B)”. However, S1B does not present any data on this.

We have corrected this.

Reviewers' Comments:

Reviewer #1 (Remarks to the Author)

I think the revised paper is a significant improvement over the original but still has some major flaws. In my opinion they satisfactorily addressed my points 1 and 4. They do not address point 5 (that their model is uninformative due to the number of free parameters) and in fact double down on it in their response to point 4 when they claim that the fact that their model accurately predicts p21 dynamics is evidence that their parameters are well approximated - this seems rather tautological to me.

The biggest problem remains with points 2 and 3 which the authors barely address at all. Regarding point 3, the paper would like to claim that p21 levels integrate DNA damage signals to inform the next cycle's G1/S progression. While they show that cells that have DNA damage also have high p21 levels and are likely to arrest in the next G1, they provide no evidence for any form of integration beyond these binary correlations. Thus, we are left with "DNA damage (natural or induced) → p53 → p21 → arrest" which is not new. Looking at the abstract it is really hard to identify what is novel about this paper. That said, it is always nice to see our understanding of the cell cycle system actually predict the results of a few experiments. In any case, the other reviewer thought the results were more informative so clearly there will be a significant fraction of the field that will find the paper to be a significant advance.

Regarding point 2, the authors are now much clearer about their findings. However, these findings remain puzzling. They report: "Strikingly, we find that whilst p21 is responsible for G1pm arrest, heterogeneity in G1 length is not p21-dependent. Because heterogeneity in G1 length is p53-dependent, this explains the correlation between p21 levels and G1 phase length; p53 activity is leading to transcriptional activation of p21, as well as causing a G1 delay. Thus, p21 is acting as a readout of p53 activity, whilst having an insignificant effect on G1 phase length. Determining the mechanisms responsible for this p53-dependent G1 delay remains an open challenge." I agree with this interpretation of their data. Their new model is that at low concentration p21 is a poor inhibitor of Cdk2 so that natural (p21-independent) variation in Cdk2 activity is what causes the variable G1 length in p21-depleted cells. But this of course still demands the question of how p53 combats this variability if it's not via p21. Based on the quoted observation it seems that p53 and not p21 ought to be on center stage.

Reviewer #2 (Remarks to the Author)

The authors have satisfactorily replied to all of the criticisms I had of the original version, and this is a significantly improved paper. It is well performed and novel and will thus make a useful contribution to the literature. I have no further criticisms, and it can be published without further revision. Since peer review should be totally transparent, I am signing my name, and I encourage the authors to make the peer review process of this manuscript public.

Daniel Fisher

Response to Reviewers

Reviewer 1

I think the revised paper is a significant improvement over the original but still has some major flaws. In my opinion they satisfactorily addressed my points 1 and 4.

They do not address point 5 (that their model is uninformative due to the number of free parameters) and in fact double down on it in their response to point 4 when they claim that the fact that their model accurately predicts p21 dynamics is evidence that their parameters are well approximated - this seems rather tautological to me.

We apologise for not addressing the reviewer's concerns regarding model accuracy. Their response to our initial submission lists only four major points and does not include a point 5 mentioned above.

We agree with the reviewer that a model's ability to inform biology is one of its most important features. However, it is now widely appreciated that the quality of a model cannot be judged solely based on the number of free parameters. In fact, it has been shown that most models in systems biology have a sloppy parameter spectrum with individual parameters being poorly constrained (Gutenkunst et al, Plos Comput Biol, 2007, doi: 10.1371/journal.pcbi.0030189). Due to the limitations of current experimental methods it is infeasible to accurately measure and constrain the parameter space in all but the simplest models. Yet, even with a poorly constrained parameter spectrum, i.e., many free parameters, models can yield well-constrained predictions (Gutenkunst et al, Plos Comput Biol, 2007). Hence, we and others argue to focus on a model's predictive power instead, in order to judge its merits.

In our manuscript, we start by simulating individual wild-type cells (Fig. 6B) and find that both the dynamics and spread of experimental p21 levels are well recapitulated (compare to Fig. 5A). In addition, the model suggests that DNA damage sustained in S-phase leads to p21 accumulation only after S-phase exit (Fig. 6C), which again is in excellent agreement with experiments (Fig. 2C). To illustrate its predictive power, we used the model to simulate the depletion of Skp2 and Cdt2, a major perturbation in parameter space that drastically changes model output (Fig. 6E). Once more, the predicted changes are in good agreement with experiments (Fig. 5A) showing, for instance, a switch-like p21 degradation upon S-phase entry in Skp2-depleted cells. Strikingly, the model also predicts anti-correlated cycles of p21 expression and DNA replication in Cdt2-depleted cells (Fig. S7E), which are present in our data as well (Fig. S6D). Taken together we thus argue that our model of p21 dynamics is indeed predictive. Moreover, since it is based on the well-established molecular network of eukaryotic cell cycle regulation, we think it is also informative as it highlights, among other things, that a bistable switch controls p21 accumulation.

We have added a sentence to stress the predictive power of our model in the final summary paragraph of the relevant Results section (p16).

Regarding point 3, the paper would like to claim that p21 levels integrate DNA damage signals to inform the next cycle's G1/S progression. While they show that cells that have DNA damage also have high p21 levels and are likely to arrest in the next G1, they provide no evidence for any form of integration beyond these binary correlations. Thus, we are left with "DNA damage (natural or induced) → p53 → p21 → arrest" which is not new. Looking at the abstract it is really hard to identify what is novel about this paper. That said, it is always nice to see our understanding of the cell cycle system actually predict the results of a few experiments. In any case, the other reviewer thought the results were more informative so clearly there will be a significant fraction of the field that will find the paper to be a significant advance.

What is important to stress is that cells with natural DNA damage have variable levels in p21. Some cells with DNA damage can continue through G1, albeit with a longer G1 phase. Thus, it is not a binary switch that DNA damage → p21 → arrest. Only if p21 accumulates to a threshold before the Restriction Point does the cell arrest in G1. Integration is a synonym of accumulation up to this threshold level, at which a switch-like decision happens (quiescence vs proliferation).

In addition, it is worth noting that we only speculate on p21 signals integrating DNA damage signals in the final paragraph of our Discussion.

Regarding point 2, the authors are now much clearer about their findings. However, these findings remain puzzling. They report: "Strikingly, we find that whilst p21 is responsible for G1pm arrest, heterogeneity in G1 length is not p21-dependent. Because heterogeneity in G1 length is p53-dependent, this explains the correlation between p21 levels and G1 phase length; p53 activity is leading to transcriptional activation of p21, as well as causing a G1 delay. Thus, p21 is acting as a readout of p53 activity, whilst having an insignificant effect on G1 phase length. Determining the mechanisms responsible for this p53-dependent G1 delay remains an open challenge." I agree with this interpretation of their data. Their new model is that at low concentration p21 is a poor inhibitor of Cdk2 so that natural (p21-independent) variation in Cdk2 activity is what causes the variable G1 length in p21-depleted cells. But this of course still demands the question of how p53 combats this variability if it's not via p21. Based on the quoted observation it seems that p53 and not p21 ought to be on center stage.

We were surprised by this result since it has been assumed that variability in G1 length would be mediated by p21 downstream of p53. However, our experimental approach convincingly shows that this is not the case and future research will determine the central players downstream of p53 in mediating variability in G1 length.

Reviewer 2

The authors have satisfactorily replied to all of the criticisms I had of the original version, and this is a significantly improved paper. It is well performed and novel and will thus make a useful contribution to the literature.

We thank Daniel Fisher for his comments and agree that peer review has improved this manuscript.